# Lactate and histone H3K18 lactylation are associated with metabolic control of gene expression in the retina

Mohita Gaur [1], Matthew J. Brooks[1], Xulong Liang[1], Ke Jiang[1¤a], Anjani Kumari[1], Milton A. English[1], Paolo Cifani[2], Maria C. Panepinto[2], Jacob Nellissery[1], Robert N. Fariss[3], Laura Campello[1¤b], Claire Marchal[1,4], Anand Swaroop[1]*

1 Neurobiology, Neurodegeneration and Repair Laboratory, National Eye Institute, National Institutes of Health, Bethesda, Maryland, United States of America, 2 Cold Spring Harbor Laboratory, Cold Spring Harbor, New York, United States of America, 3 Biological Imaging Core, National Eye Institute, National Institutes of Health, Bethesda, Maryland, United States of America, 4 In silichrom Ltd, Newbury, United Kingdom

¤a Current address: Cleveland Clinic Foundation, Cleveland, Ohio, United States of America
¤b Current address: Avista Therapeutics, Pittsburgh,United States of America
☯ These authors contributed equally to this work.
* swaroopa@nei.nih.gov

## Abstract

High aerobic glycolysis in retinal photoreceptors, as in cancer cells, is implicated in mitigating energy and metabolic demands. Lactate, a product of glycolysis, can exert epigenetic regulation through histone lactylation in cancer. Here, we show that enhanced ATP production during mouse retinal development is achieved primarily through increase in glycolysis. Histone lactylation, especially H3K18La, parallels increased glycolysis and lactate levels in the developing retina. Multi-omics analyses, combined with confocal imaging, reveal the localization of H3K18La near H3K27Ac in the euchromatin at promoters of active retinal genes. In mouse retinal explants, glucose metabolism is associated with lactate levels as well as H3K18La and consequently gene expression. However, inhibition of glycolysis with 2-deoxyglucose (2-DG) reduces global H3K18La and H3K27Ac marks with somewhat distinct transcriptional changes. Evaluation of accessible chromatin at H3K18La-marked promoters uncovers an enrichment of GC-rich motifs for transcription factors of SP, KMT and KLF families, among others, indicating the specificity of H3K18La-mediated gene regulation. Our results indicate glycolysis/lactate/H3K18La as a potential axis for transcriptional response to changing metabolic conditions in the retina, especially photoreceptors.

purpose. The work is made available under the Creative Commons CC0 public domain dedication.

**Data availability statement:** Data and code availability • The next generation sequencing data generated in this study are available at the Gene Expression Omnibus (GEO; accession number GSE291677). https://www.ncbi.nlm.nih.gov/geo/query/acc.cgi?acc=GSE291677 • Custom code used for data analysis is available in GitHub (https://github.com/NEI-NNRL/2026_Mouse_Retina_H3K18La) Materials availability All unique/stable reagents generated in this study are available with a completed Materials Transfer Agreement per NIH policy. Further information and requests for resources and materials should be directed to and will be fulfilled by the corresponding author, Anand Swaroop (swaroopa@nei.nih.gov).

**Funding:** This research support and salaries of NIH researchers were from Intramural Research Program of the National Eye Institute, National Institutes of Health, Grants ZIAEY000450 and ZIAEY000546 (to A.S.). Cancer Center Support Grant 5P30CA045508 provided support for CSHL mass spectrometry shared resource (to M.C.P.). The funders had no role in study design, data collection and analysis, decision to publish, or preparation of the manuscript. The contributions of the NIH author(s) are considered Works of the United States Government. The findings and conclusions presented in this paper are those of the author(s) and do not necessarily reflect the views of the NIH or the U.S. Department of Health and Human Services.

**Competing interests:** The authors have declared that no competing interests exist.

## Author summary

Retina relies on aerobic glycolysis to sustain visual function, but how metabolic state is communicated to the chromatin landscape remains poorly understood. Here, we identify histone H3 lysine 18 lactylation (H3K18La) as a glycolysis- and lactate-sensitive histone mark in the retina. Using *in vivo* developmental profiling and glucose-supplemented retinal explants, we show that increased glycolytic flux is concordant with enhanced H3K18La marks at *cis*-regulatory elements in photoreceptor genes. Conversely, glycolytic blockade with 2-deoxyglucose in mouse retinal explants leads to global loss of H3K18La and H3K27Ac with partially overlapping but distinct transcriptional consequences. Our findings demonstrate that H3K18La links glycolysis and lactate to chromatin state and gene expression. We suggest that differential regulation of H3K18La and H3K27Ac marks can coordinate the response to metabolic conditions and fine-tune retinal photoreceptor gene expression and function.

## Introduction

Reversible epigenetic modifications of histones impart an adaptable metabolic control on gene expression, integrating developmental cues and extrinsic signals into stable transcriptional programs [1]. Among these, the most extensively studied modification is probably histone acetylation, which promotes accessibility of chromatin and activates transcription by recruiting bromodomain-containing proteins and co-activators [1,2]. In the retina, acetylation marks such as H3K27Ac have been tightly linked to enhancer activation and induce expression of genes required for photoreceptor differentiation and function [3,4].

Recent advances in mass spectrometry have expanded the current catalog of histone post-translational modifications (PTMs) that now include an array of metabolite-derived acylations, such as crotonylation, succinylation, and lactylation [5–7]. Metabolic intermediates, produced primarily from glycolysis and oxidative phosphorylation (OXPHOS), serve as key modifiers of histone PTMs [8,9]; these include acetyl-CoA and acyl Co-A derivatives that alter lysine residues [5,10]. Though acetylation is broadly associated with transcriptionally active regions, histone lactylation has emerged as a context-specific change providing a direct biochemical link between glucose metabolism and chromatin regulation [7]. First identified in cancer cells, histone lactylation (notably H3K18La) was reported to function as a metabolic sensor controlling macrophage polarization and tumor progression [7]. Lactylation of histones has since been implicated in immune evasion [11], cardiac stress response [12] and ischemia-reperfusion injury [13], highlighting its divergent functions in physiology and disease. Additionally, histone lactylation contributes to embryonic development and lineage specification [14] as well as long-term neuronal activity and memory formation [15].

The mammalian retina, a highly metabolic yet terminally differentiated non-proliferative tissue, exhibits an unusually high reliance on aerobic glycolysis [16],

which generates energy as well as biosynthetic precursors to support visual function [17,18]. The retina consists of an array of diverse and highly specialized neurons that are stratified in three cellular layers for photon capture and processing of visual information [19]. The rod and cone photoreceptors account for almost 75% of the cells in most mammalian retina and initiate the process of vision by converting photons to electrical signals through the process of phototransduction [20]. The human retina includes over 105 million densely packed rods and about 6 million cones that permit vision in dim and daylight, respectively, by closing the cGMP-gated channels in response to photon capture [21]. Maintenance of physiological state as well as daily renewal of membranous discs place extensive energy and anabolic demands on the retinal photoreceptors [22–26].

Glucose is the primary energy source in the neural retina, with photoreceptors being the principal consumer [27]. Aerobic glycolysis in photoreceptors maintains high anabolic activity and survival [18,28]. Glucose uptake in photoreceptors depends on the retinal pigment epithelium (RPE) acting as a barrier to the choroidal blood vessels while concurrently ensuring a steady supply of glucose, oxygen and nutrients. The excess lactate produced by glycolysis in photoreceptors is transported to the RPE, where it serves as the substrate for OXPHOS and suppresses glucose utilization in the RPE [29]. This complementary relationship between photoreceptors and RPE is suggested to create a metabolic ecosystem [30–32].

The lactate levels in the mammalian retina are 5–10 times higher than other tissues [33]. Specifically, the retinal photoreceptors exhibit prominent expression of lactate dehydrogenase A (LDHA) that converts pyruvate to lactate [26]. More recently, glycolysis and enhanced lactate levels have been implicated in mediating morphogenesis as well as transcriptional patterns in developing eye organoids [34]. Genetic alterations that affect glycolysis can also influence the structure of photoreceptor outer segment [18]. Additionally, metabolic underpinnings of photoreceptor dysfunction are correlated to glucose uptake and metabolism in retinal disorders [28,35]. These studies point to broader regulatory functions of high lactate levels. We currently have poor understanding of how retina and the photoreceptors adapt to changing energy and metabolic demands during development, aging and disease.

In this report, we directly demonstrate the concordance of augmented histone H3K18La with high lactate and glycolytic flux in developing mouse retina and under high glucose conditions in explant cultures of mouse retina. Predictably, inhibition of glycolysis leads to a reduction in lactate level as well as a global loss of H3K18La. Furthermore, H3K18La peaks near the promoter regions partially colocalize with H3K27Ac in the euchromatin and are associated with higher expression of genes involved in retinal development and/or photoreceptor function. We also identify transcription factor (TF) binding motifs, including those of Krüppel-like factor (KLF) and specificity protein (SP) families, at the accessible H3K18La peaks. Our results implicate aerobic glycolysis and lactate levels, and consequently augmented histone H3 lactylation at K18 residue, as significant contributors to transcriptional adaptation in the retina, and especially in the photoreceptors.

## Results

### Enhanced glycolytic flux contributes to lactate accumulation in developing retina

We used Seahorse XFe24 Analyzer to measure glycolysis (extracellular acidification rate or GlycoECAR) and mitochondrial respiration (oxygen consumption rate or MitoOCR) of developing and mature mouse retina (Fig 1A-1C). We noted that GlycoECAR continues to increase after postnatal day 6 (P6) (Fig 1B), whereas MitoOCR is only marginally higher in mature (P28) retina (Fig 1C). Total ATP production, contributed from both glycolysis and mitochondrial respiration, was significantly enhanced as development proceeds (Fig 1D). ATP production contributed by OXPHOS exhibited little change from P2 to P28. Until P10, mitochondrial respiration appeared to serve as the primary source of ATP, with glycolysis being a minor contributor. However, glycolytic flux continued to increase from P6 onward and becomes the major energy source after P14, as retinal neurons terminally differentiate, form synapses and become functionally mature (Fig 1D and 1E). During retina development, quantifications using bioluminescent Lactate-Glo assay demonstrated a steady increase in lactate levels (Fig 1F). Significantly higher lactate levels were detected in retinas at late developmental stages (P14 and

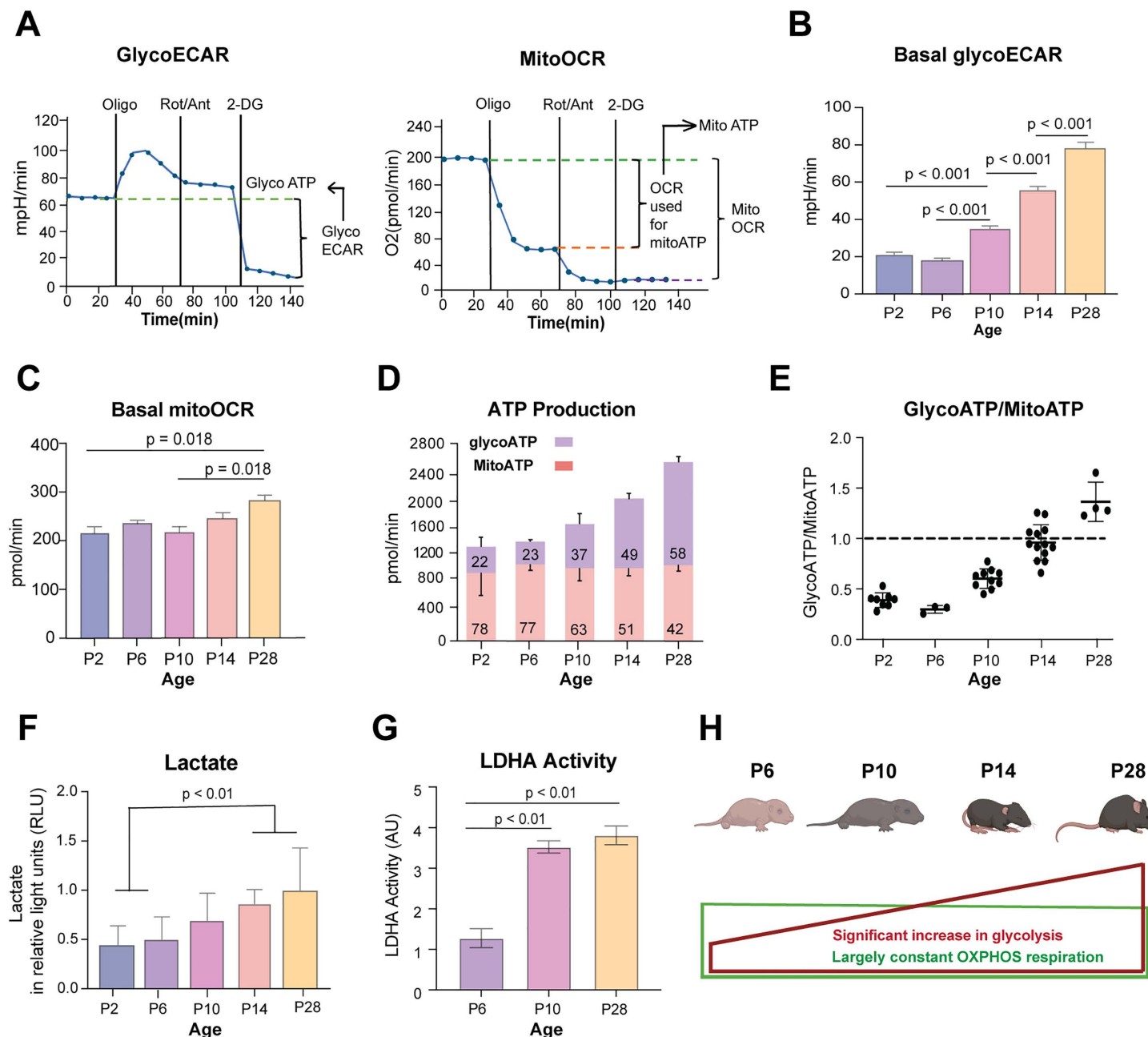

**Fig 1. Contribution of aerobic glycolysis and lactate to ATP generation during retinal development.** A) Graphical representation of glycolytic flux profile (left) and mitochondrial respiration profile (right) using Seahorse XFe24 Analyzer. The left graph represents extracellular acidification rate (ECAR), which is largely proportion to glycolytic activity of cells and the right graph represents oxygen consumption rate (OCR), which is an indicator for mitochondrial respiration. The X-axis shows duration of the assay, and the timing of sequential inhibitor additions are indicated by vertical lines. The initial baseline (0–36 min) represents normal glycolysis level and respiration level before any metabolic interventions. With Oligomycin (an inhibitor of mitochondrial complex V), oxygen consumption drops drastically whereas glycolytic flux displays a sharp increase followed by a slight decline. The addition of Rotenone/Antimycin A (inhibitors to complex I and III) further shuts down mitochondrial respiration but with minimal influence on glycolysis flux. The final addition of the glycolytic inhibitor 2-Deoxy-D-glucose (2-DG) completely abrogates glycolysis. B) and C) Basal GlycoECAR and basal mitoOCR were quantified from Seahorse assays using retinal punches isolated from mice at different developmental age groups (n = 13–35 retinal punches per age group from 4 mice). Data are presented to compare the metabolic activity between age groups, with glycoECAR reflecting glycolytic activity and mitoOCR reflecting mitochondrial respiration. D) ATP production rates calculated from collated MitoOCR and GlycoECAR data from B and C. The number on the bar indicates percentage. E) Ratio of glycolysis derived ATP to mitochondrial derived ATP was calculated from data collated for D. F) Lactate concentration was quantified for different age groups in retina tissue samples using Lactate-Glo Luciferase Assay (Promega, J5021), n = 4 except for P28,

n = 3. Data plotted as mean luminescence values (relative light units – RLU) from each replicate per age group. G) Lactate Dehydrogenase (LDH) Assay Kit (ab102526, Abcam) was used to measure LDH activity in the mouse retina from different age groups (P6, P10, P28; n = 4 per age group). H) Pictorial representation of ATP production during retinal development, showing enhanced glycolysis derived ATP production (represented as red rectangle), however proportional percentage contribution from mitochondrial derived ATP remains largely constant with age (represented as green rectangle). Fig 1H Created in BioRender. https://BioRender.com/cxlfw0l. All data in the figures are presented as mean ± SEM. Statistical significance is indicated by p-values displayed above the bars. One-way ANOVA with Tukey's post hoc comparison was used to determine the significance, *, p < 0.05. Abbreviations: ECAR, extracellular acidification rate; OCR, oxygen consumption rate; OXPHOS, oxidative phosphorylation.

P28) when compared to early developmental stages (P2 and P6) and correlated with enhanced glycolysis (Fig 1B). We also detected a significant increase in lactate dehydrogenase activity from P6 to P10 and P28 (Fig 1G). Overall, comprehensive GlycoECAR and mitoOCR measurements indicated augmented aerobic glycolysis in developing retina, presumably to meet higher energy and metabolic demands (Fig 1H).

**H3K18La in the retina correlates with enhanced glycolysis**

We hypothesized that augmented aerobic glycolysis and high lactate levels in the retina contribute to the chromatin state and gene expression via histone lactylation (Fig 2A). To test this hypothesis, we first performed high-resolution mass spectrometry analysis of histones from the P28 mouse retina and detected multiple lactylation sites across histone proteins (Fig 2B and S1 Table). A representative MS/MS spectrum for the peptide from H3.3 containing KLa at position 18, one of the most robust identifications, is shown in S1A Fig. We then performed immunoblot analysis using antibodies against H3K18La and H4K12La since these two histone lysine lactylation sites are reported to link glucose metabolism to epigenetic regulation [7,36]. We observed a significant increase in H3K18La levels during retina development (from P2 to P28), but no significant change was evident in Pan-lactyl or H4K12La signals (Fig 2C).

Immunohistochemistry (IHC) of developing mouse retina sections using anti-H3K18La and Pan-lactyl antibodies revealed prominent H3K18La and Pan-lactyl labeling in the nuclei as well as axonal and dendritic processes of retinal neurons (Fig 2D). The fluorescence in the outer nuclear layer, which contains the nuclei of photoreceptors, appeared somewhat diffused in early postnatal development but gradually acquired a more defined pattern that was consistent with unique euchromatin architecture of mature photoreceptors (Fig 2D). H3K18La immunoreactivity was detected across multiple retinal layers but showed the most robust and developmentally regulated increases in the outer nuclear layer (Fig 2D). The signal in the inner retina likely reflects basal levels of lactylation in interneurons or glia. H3K18 lactylation therefore was present more broadly in retinal cells but exhibited a prominent augmentation in photoreceptors during maturation. Our results suggest a close correlation of H3K18La levels with glycolysis in developing and mature retinal photoreceptors in contrast to Pan-lactyl immunostaining that is broadly enhanced in the nuclei of most retinal cell types (Fig 2D).

**Chromatin context-dependent gene regulation by H3K18La**

We then investigated genome-wide occupancy of H3K18La using CUT&Tag assay, during postnatal mouse retinal development (P4, P10 and P28) and asked whether histone lactylation is linked to functional maturation of the retina. Consensus peaks were determined by using q-value threshold of 1x10-6 and selecting those observed in at least two (out of 3) replicates. Early postnatal retina (P4) displayed numerous small and dispersed H3K18La peaks (51,085), which were reduced at P10 (30,777). By P28, we observed a greater number of defined peaks (64,973) (Figs 3A and S1B). Interestingly, the number of promoters with a H3K18La peak exhibited a dramatic increase in the adult retina (P28) with higher distribution proximal to TSS, even though a majority of H3K18La peaks were detected in gene bodies (Fig 3B and 3C). Peaks being called at P4 appeared to be widely distributed and lower in signal intensity prior to coalescing to fewer, but more robust peaks at P10, and before finally establishing the highest peak signals at P28 (S1B Fig). Select examples of genes involved in lactate production (*Aldoa* and *Ldha*) and phototransduction (rhodopsin, *Rho*) demonstrated the dynamics of H3K18La binding during development (Fig 3D). Gene ontology (GO) enrichment analyses of genes associated with

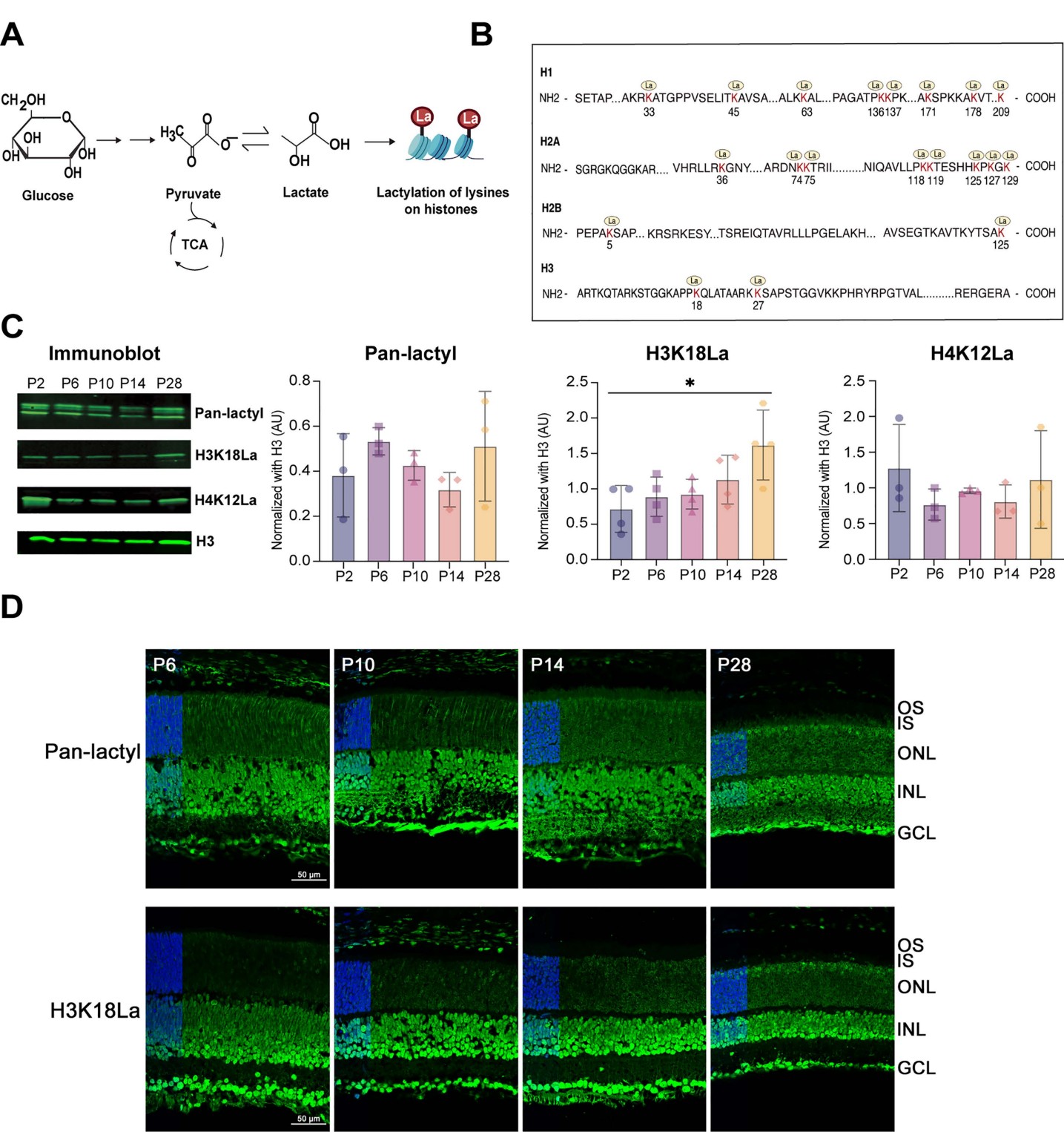

**Fig 2. Histone lactylation in the retina.** A) Graphical illustration of non-metabolic role of high lactate production in retinal photoreceptors cells because of Warburg's effect. The lactate produced can act as chemical tag "La" for histone which further modulates gene transcription in retina. Fig 2A Created in BioRender. https://BioRender.com/gsgb8sv B) Illustration of the number of unique lactylated (La) lysines (K) identified per histone by mass spectrometry in derivatized tryptic-digested peptides from adult (P28) mouse retina. The identified lactylation sites on lysine residues are highlighted in red and

marked by yellow circle icons with "La" written inside. C) Immunoblot-based quantification of histone KLa in purified histone samples at different ages of mice for Pan-lactyl (n = 3), H3K18La (n = 4) and H4K12La (n = 3) with the experiment repeated twice. ImageStudio software was used for quantification. Statistical significance was analyzed using one-way ANOVA with post hoc Tukey's test. Immunoblots for Pan-lactyl, H3K18La, and H4K12La show molecular weights of 15 kDa. All data are presented as mean±SEM. *, p < 0.05. D) Immunohistochemical analysis of Pan-KLa or H3K18La in retina tissue samples from different age groups of mice. Inset represents a zoomed-out view with DAPI staining in the ONL on the left. Abbreviations: OS, outer segment; IS, inner segment; ONL, outer nuclear layer; INL, inner nuclear layer; GCL, ganglion cell layer. Scale bar = 50 μm.

H3K18La peaks in promoters, enhancers or gene bodies showed a differential enrichment of cellular pathways (S1C Fig and S2-S4 Tables). H3K18La peaks were enriched in promoters of genes involved in canonical Wnt signaling, synaptic membrane adhesion and circadian regulation, whereas those in the enhancer regions exhibited dominance of retinal functions including eye development, visual perception and axon guidance. The latter observation is consistent with a higher correlation of enhancer elements with cell-type-specific genes compared to the promoters [36]. The H3K18La peaks in gene bodies were enriched for ERK1/2 cascade, synapse assembly, Wnt signaling and photoreceptor cell maintenance.

We further evaluated chromatin annotation of the peaks using a ChromHMM model derived from previously published retina data [3] (S2A Fig). H3K18La marks showed an enrichment for active promoter annotations with strong TSS-proximal enhancer annotations, which became more prominent at P10 and P28. H3K18La peaks at enhancers and gene bodies revealed similar patterns of chromatin annotations, including enhancer and transcription permissive chromatin.

To gain additional insights, we performed colocalization of H3K18La peaks with published retina ATAC-seq, H3K27Ac [3] and CUT&RUN genome occupancy data for the key photoreceptor transcription factor NRL [38], which is essential for rod differentiation [39]. We focused our analysis on region-specific overlaps, to directly address whether H3K18La preferentially co-localizes with H3K27Ac at defined regulatory elements. Our results showed that H3K18La peaks exhibited high co-localization percentage with H3K27Ac and accessible chromatin at promoters but much lower at gene bodies or enhancers (Figs 3E and S2B). Confocal microscopy results revealed better overlap of H3K18La with H3K27Ac (ROI colocalized - 85.38%) and NRL (ROI colocalized - 82.57%) compared to the repressive mark H3K27me3 (ROI colocalized - 54.19%) (Figs 3F and S2C). Together, these results indicate an enrichment of H3K18La at euchromatic promoters in photoreceptors with overlaps with H3K27Ac and NRL while remaining distinct from the repressive mark H3K27me3.

Assessment of promoters of protein-coding genes that are marked by H3K18La during retinal development (Fig 3G) revealed co-occurrence of H3K18La with previously reported active histone marks, H3K27Ac and/or H3K4me3 [3]. Notably, co-occurrence of these marks at promoters was only 38% at P4 (early development) but increased to 84.4% in P28 (mature) retina. Predictably, the corresponding genes having all three active promoter marks reflect an enrichment of biological pathways relevant to retinal development (Figs 3H, 3I and S2D and S5 Table); these include Wnt signaling, neurogenesis and eye development (Crx and Gsk3b). Acquired H3K18La peaks at P10 are enriched for pathways associated with chromatin (Kdm5b), mitochondria and cilium organization (Bbs4) and at P28 with transcription initiation (Tbp), mRNA processing and mitochondrial gene expression (Taco1). These results are consistent with high biosynthetic and energetic demands of mature photoreceptors and indicated that H3K18La acquisition is developmentally staged and adapts to the retina's changing biological requirements.

We then evaluated co-occurrence of H3K18La binding in the retina with those in other tissues of differing metabolic activity [37]. After processing the data in an identical manner (see methods), we tallied and compared the presence of H3K18La peaks in promoters of protein coding genes (Figs 3J and S2E). Notably, approximately one third of peaks are specific to the retina with a significant enrichment of retina-specific genes. The top enriched pathways are sensory perception and modulation of chemical synaptic transmission, underscoring the role of H3K18La in tissue-specific transcriptional programs (S6 Table).

## H3K18La couples the metabolic state with gene regulation in developing retina

To determine whether promoter-associated H3K18La is linked to transcriptional output, we integrated the CUT&Tag data with RNA-seq from developing retina. We first quantified 108,253 merged consensus peaks for the three ages

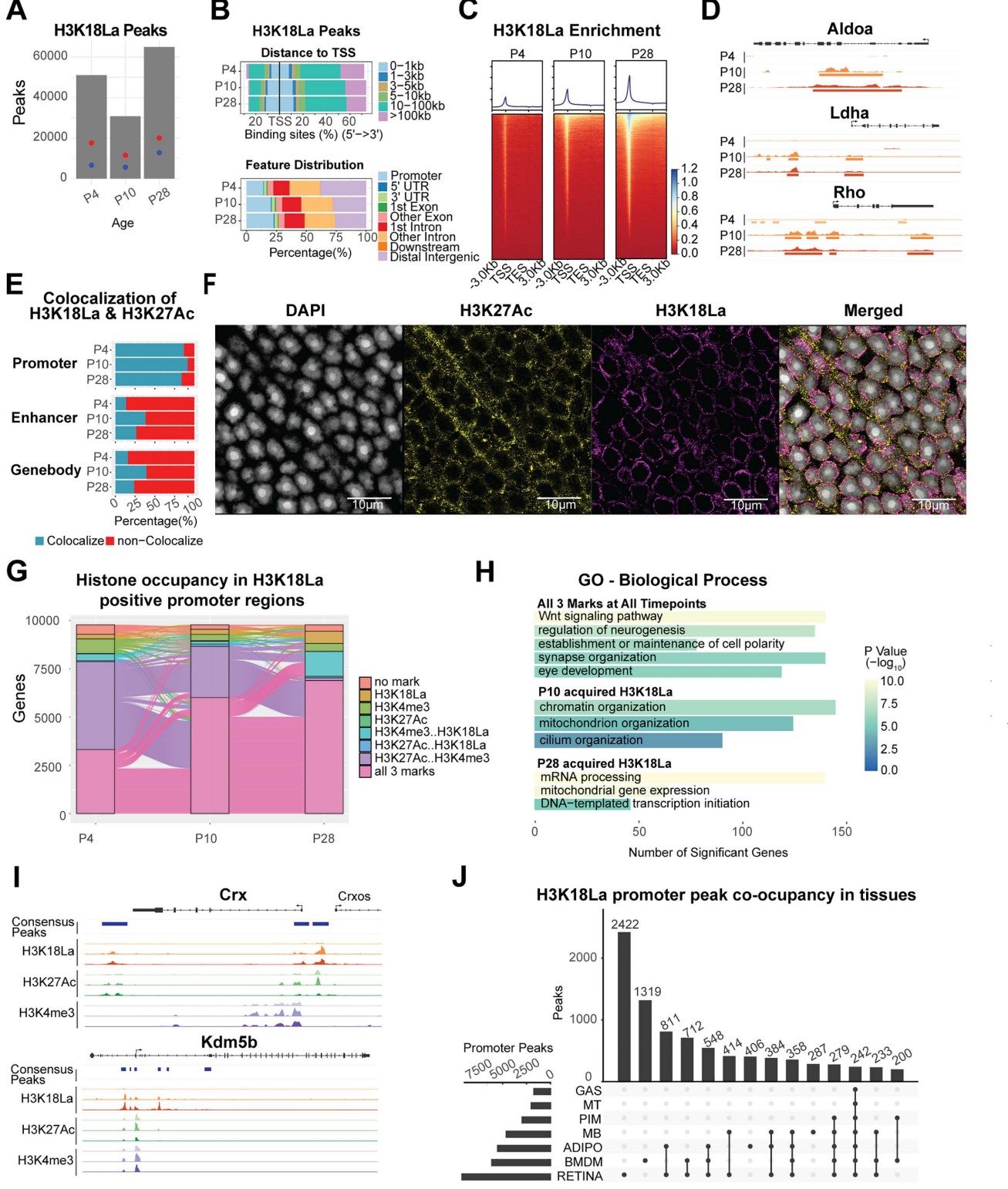

**Fig 3. Genome-wide profiling of H3K18La.** (A) The number of consensus peaks passing a 1x10-6 FDR for each replicate per post-natal timepoint. The red dot indicates the number of genes containing a peak (17,647, 11,511, and 20,087 respectively), whereas the blue dot represents the number of genes containing a peak in the proximal promoter (6,646, 5,627, and 12,822 respectively). (B) Upper panel: Percentage of H3K18La bound consensus

peak loci distance relative to TSS. Lower panel: Percentage of H3K18La bound consensus peak loci to gene annotation feature. (C) Heatmap of H3K18La-bound peak signal enrichment for all genes and their flanking 3kb region. (D) Genomic histogram traces of H3K18La at each timepoint for genes involved in glycolysis (*Aldoa* and *Ldha*) and the phototransduction cascade in rod photoreceptors (*Rho*). The histogram traces are group scaled for each gene. The bars under each histogram represent consensus peaks. (E) Colocalization of H3K18La peaks with H3K27Ac [3] bound regions. (F) Confocal immunofluorescent images showing colocalization of H3K27Ac and H3K18La in nuclear periphery of photoceptor cells. (G) Dynamic active histone mark (H3K4me3 and H3K27Ac [3]) co-occupancy with H3K18La in the proximal promoter of genes containing H3K18La during development. (H) Selection of GO Biological Process gene sets enriched for genes from panel (G) containing all three histone marks at all timepoints or genes that acquired H3K18La loci at P10 or P28. (I) Genomic histogram traces of histone marks during development for representative selected genes found in panel H. H3K18La marks at P4, P10, and P28 (Red), H3K27Ac [3] at P3, P10, P21 (Green), and H3K4me3 at P3, P10, and P21 (Purple). (J) The UpSet plot illustrates the differential H3K18La co-occupancy at promoter regions of protein coding genes between retina and other metabolically distinct tissues. Single black dots represent promoter peaks unique to one tissue. Connected black dots represent promoter peaks shared among the indicated tissues. Thus, the y-axis quantifies the number of H3K18La-marked promoters corresponding to each specific intersection (i.e., unique or shared across tissues) [37]. Abbreviations: TSS, transcriptional start site; TES, transcriptional end site; GO, gene ontology; GAS, gastrocnemius; MT, post-mitotic end-state myotubes; PIM, post-ischemia macrophages; MB, myoblasts; ADIPO, adipose tissues; BMDM, bone marrow-derived macrophages.

(P4, P10 and P28) and visualized the data by Principal Component Analysis (PCA) (Fig 4A, upper panel) together with the published retinal gene expression data [40] (Fig 4A, lower panel). PC1 demonstrated the largest variance, which can be attributed to age. At P4 and P28, the replicates were well clustered and revealed biological differences between the two age groups. In contrast, the P10 data showed variability among replicates, likely indicative of dynamic acquisition of H3K18La during an active differentiation state of morphogenesis and synapse formation.

Differential binding (DB) analysis of the merged and quantified consensus peaks identified 34,794 peaks showing significant DB (FDR < 0.01) during development (S7 Table). Furthermore, only 126 of the 5629 DB peaks in promoters of 5170 protein-coding genes (Fig 4B) exhibited a decrease in binding from P4 to P28. Pearson correlation analysis of 5629 significant DB peaks in promoters of protein-coding genes uncovered 1734 genes that demonstrated a positive correlation of greater than 0.8 between H3K18La DB peak quantitation and RNAseq data from developing mouse retina (Fig 4C, left panel), and 1253 genes revealed a negative correlation of less than -0.8 (Figs 4C, right panel and S2F). Enrichment analysis of the positively correlated genes identified the pathways involved in phototransduction, synapse, and glycolysis (S2F Fig and S8 Table). Thus, promoter H3K18La seems to have a distinct role depending upon the chromatin context.

We then focused on DB promoters from 1,734 positively correlated genes to explore H3K18La in a broader chromatin context. Most of these promoters (57.5%) possess active histone marks, H3K27Ac or H3K4me3, at P4 prior to acquiring H3K18La and exhibited an increase in gene expression during development (Fig 4D). At P4, only 34.6% of positively correlated promoters had all three modifications (H3K27Ac, H3K4me3, H3K18La), whereas 83.9% carried all three marks by P28 (Fig 4D). These genes are enriched for response to peptide hormone, histone deacetylation, and eye development (Fig 4E, left panel and S9 Table) and include *Akt3* and *Rorb*, which contained all three histone marks from P4 onwards. *Akt3* (Fig 4F, top left panel) is important for regulating the canonical insulin/ATK/mTOR pathway and reprogramming photoreceptor metabolism and survival [41]. *Rorb* (Fig 4F, bottom left panel) is a nuclear receptor involved in rod photoreceptor development [42]. A robust H3K18La mark is observed at P10 in genes associated with photoreceptor maturation including visual perception and synapse processes (Fig 4E, right panel). As an example, *Reep6* (Fig 4F, top right panel) plays a critical role in trafficking guanylate cyclases in rod photoreceptors, and its loss results in retinopathies [43,44]. Neuronal synaptobrevin (*Vamp2*) (Fig 4F, bottom right panel) facilitates fusion of synaptic vesicles [45]. Thus, H3K18La seems to be associated with establishment of specialized neuronal functions by activating transcription from promoters of genes with a permissive chromatin context. The presence of H3K18La at P10 also indicated enhanced glycolytic activity and attainment of glucose-dependent processes necessary for effective visual function. Together, these results suggest that H3K18La is gained at promoters already in a permissive chromatin state and associated with developmental and neuronal programs involved in retinal maturation.

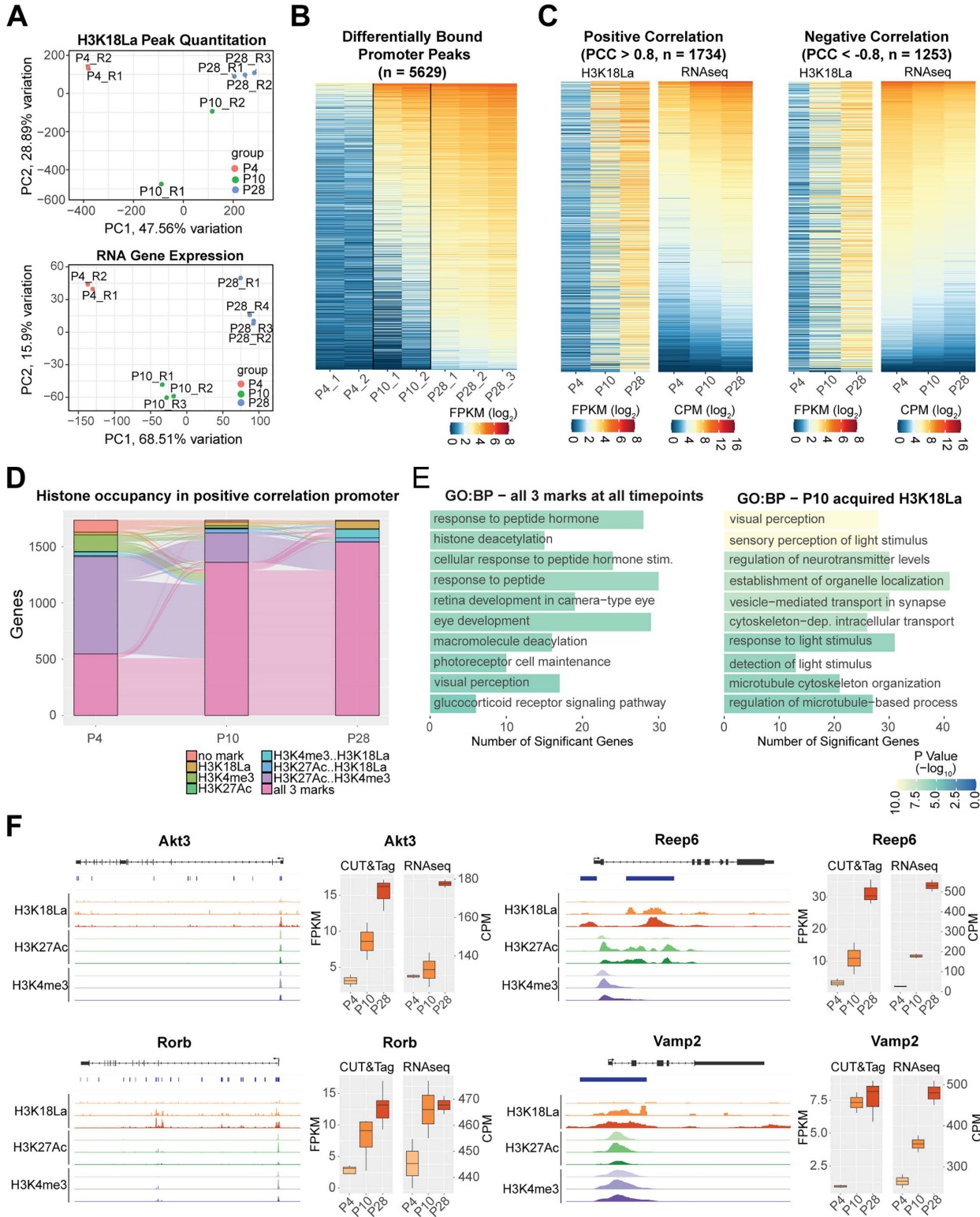

**Fig 4. Differential binding of H3K18La during development.** (A) Upper panel: Principal component analysis of quantitative peak binding for all H3K18La samples. Lower panel: Principal component analysis of RNA expression [38] for the same genes having H3K18La peaks. This showed the reproducibility of biological replicates in the datasets. (B) Heatmap of H3K18La quantitative binding for differentially bound peaks in the promoter region

of protein coding genes. (C) Left panel: Positive correlation of quantitative peaks of H3K18La found in panel (B) with retina RNA-seq expression [38]. Right panel: Negative correlation of quantitative peaks of H3K18La found in panel (B) with retina RNA-seq expression [38]. (D) Dynamic active histone mark (H3K4me3 and H3K27Ac [3]) co-occupancy with H3K18La in the positive correlation set of genes from panel (C). (E) Selection of GO Biological Process gene sets enriched for genes from panel (D) containing all three histone marks at all timepoints or genes that acquired H3K18La loci at P10. (F) Left: Genomic histogram traces of histone marks during development for representative selected genes found in panel (E). H3K18La marks at P4, P10, and P28 (Red), H3K27Ac at P3, P10, P21 (Green), and H3K4me3 at P3, P10, and P21 (Purple). Right: Boxplot of quantitative binding and RNA-seq expression and CUT&Tag analysis. Abbreviations: PCA, principal component analysis; PC1, principal component 1; PC2, principal component 2; Pearson correlation coefficient (PCC); FPKM, fragments per kilobase per million reads; CPM; counts per million; GO, gene ontology; BP, biological process.

## Regulation of histone lactylation by glucose metabolism

To directly examine the relationship of histone lactylation with glycolysis, we performed a glucose modulation experiment in explant cultures of P28 mouse retinas. We exposed retinal explants to two different conditions: elevated glucose levels (5 mM to 25 mM) and addition of 10–20 mM 2-deoxy-D-glucose (2-DG), non-metabolizable glucose analog that inhibits glycolysis. We then measured the amount of lactate secreted in the media, together with Pan-lactyl, H3K18La and H3K27Ac changes in the retinal tissue (Fig 5A). We chose H3K27Ac as a representative acetylation mark associated with transcriptional activation. Both lactate production and histone lactylation levels (H3K18La and Pan-lactyl; n = 3, p < 0.05) were induced by glucose in a dose-dependent manner (Fig 5B and 5C); however, incubation with 2-DG significantly decreased both lactate and histone KLa levels (Pan-lactyl p < 0.01; H3K18La p < 0.05, n = 3) (Fig 5D and 5E). High glucose conditions did not significantly alter H3K27Ac levels in retinal explants (Fig 5C); yet the treatment with 2-DG, which competed with glucose for binding, significantly reduced the H3K27Ac levels (p < 0.05) (Fig 5E). Thus, lactate levels and consequently histone lactylation are dependent on glycolytic flux in retinal explants. Our findings reflect changes in metabolically coupled epigenetic state rather than photoreceptor survival outcomes.

## Glucose availability and glycolytic inhibition by 2-DG differentially affect H3K18La dynamics

To evaluate whether glucose availability regulates gene expression in the retina, we cultured the retinal explants under low (5 mM) and high (25 mM) glucose conditions and performed CUT&Tag for H3K18La and H3K27Ac together with RNA-seq. This approach allowed us to examine how glycolytic flux influences promoter occupancy of H3K18La compared to H3K27Ac and assess downstream transcriptional responses. We observed a higher number of H3K18La peaks and peaks within promoters at 25 mM glucose compared to 5 mM (S3A Fig), but with relatively little change in the total number of genes (i.e., 27,663 in 5 mM compared to 27,183 in 25 mM). On the other hand, the total number of H3K27Ac peaks (24,921 and 17,273 in 5 mM and 25 mM, respectively) and those detected within genes and their promoters (12,918 and 11,369 in 5 mM and 25 mM, respectively) decreased under 25 mM glucose conditions. Notably, the number of peaks and their location near transcription start site (TSS) increased with 25 mM glucose condition for both histone marks (Figs 6A and S3B). For H3K27Ac, our analysis showed a paradoxical pattern: fewer unique promoters were marked overall, yet those that were marked exhibited a higher density of peaks, indicating that elevated glucose leads to an increased number of H3K27Ac peaks per promoter. PCA of the quantified peaks under 5- and 25-mM glucose conditions demonstrated a larger separation of peaks for H3K18La compared to H3K27Ac (S3C Fig). Visual inspection of the H3K18La and H3K27Ac peaks, and their read pileups (S3D Fig), indicated an overall increase in signal intensity, presence of additional peaks, and merging of smaller peaks into larger robust peaks with glucose supplementation (i.e., at 25 mM). Promoters of protein coding genes that gained H3K18La during development appeared to gain both H3K18La and H3K27Ac marks under high glucose condition (Fig 6B).

   Gene expression changes in retinal explants were then compared to corresponding changes in H3K18La peaks. A majority of the significantly differentially expressed (FC > 2, FDR < 1%) genes showed an increase in expression with higher glucose (Fig 6C and S10 Table); these include genes involved in retina development (*Neurod1*, *Casz1*, and *Crxos*)

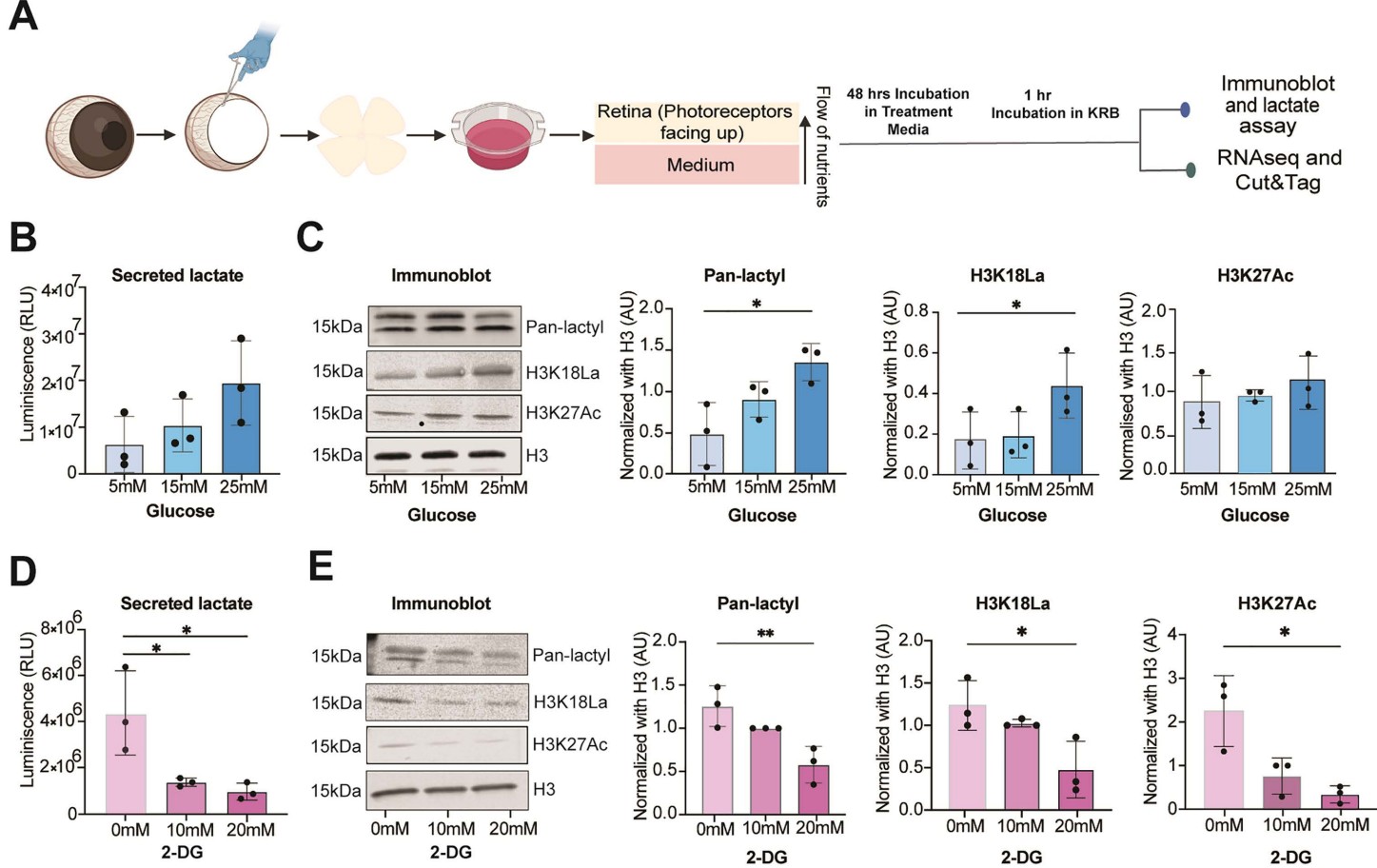

**Fig 5. Glycolysis regulates histone lactylation in the retina.** (A) Representation of the retinal explant workflow. Key steps in the experiment are shown. The isolated retina was cultured in explant media for 48 hours before proceeding for subsequent experiments as outlined in the Fig 5A Created in BioRender. https://BioRender.com/4sbqcn9. (B, D) Extracellular lactate levels using Promega Lactate-Glo Luciferase Assay (Promega, J5021) and (C, E) histone lactylation (KLa) levels using immunoblot were measured in retinal explants cultured for 48 hours in different concentrations of glucose (5 mM, 15 mM, and 25 mM) or 2-Deoxy-D-glucose (2-DG) (10 mM and 20 mM) with 5 mM glucose as a baseline control in 2-DG experiments. Extracellular lactate, the amount of lactate secreted into the culture media, was quantified and the amount of lactate was correlated with luminescence levels, expressed in relative light units (RLU) on Y axis as shown. For (C, E) immunoblots, histones were extracted from retinal tissues cultured under the same conditions for 48 hours. Pan anti-KLa, H3K18La, and H3K27Ac antibodies detected protein bands at approximately 15 kDa. Data are presented as mean ± SEM (n = 3 biological replicates, *, p < 0.05; **, p < 0.01). Statistical significance was determined using one-way ANOVA followed by Sidak's multiple comparisons test. Abbreviations: RLU, relative light units; AU, arbitrary unit; KRB, krebs buffer; 2-DG, 2-Deoxy-D-glucose; H3, Histone 3.

and those associated with retinal disease (*Impg1/2*, *Gucy2f*, *Mpp4*, *Arl6*, and *Rp1*). Additionally, genes involved in metabolic pathways and rod phototransduction demonstrated an increase in H3K18La and H3K27Ac marks under high glucose conditions (Figs 6D and S3E). Our analyses of developing retina and glucose-supplemented explants suggest that augmented glycolysis is concordant with increased H3K18La marks indicating a tight linkage of H3K18La to glycolysis. To investigate how glycolytic blockade affects transcriptional output, we cultured retinal explants in the presence of 20 mM 2-DG. CUT&Tag profiling revealed a global reduction in H3K18La and H3K27Ac peaks compared to control glucose conditions (S4A Fig). Notably, we detected a pronounced reduction in the number of promoter peaks for H3K18La (Fig 6E); yet, a concomitant reduction is not evident in H3K27Ac promoter peaks. PCA analysis of promoter peak quantitation further demonstrated a clear separation between 2-DG and control samples (S4C Fig). Glycolytic inhibition by 2-DG

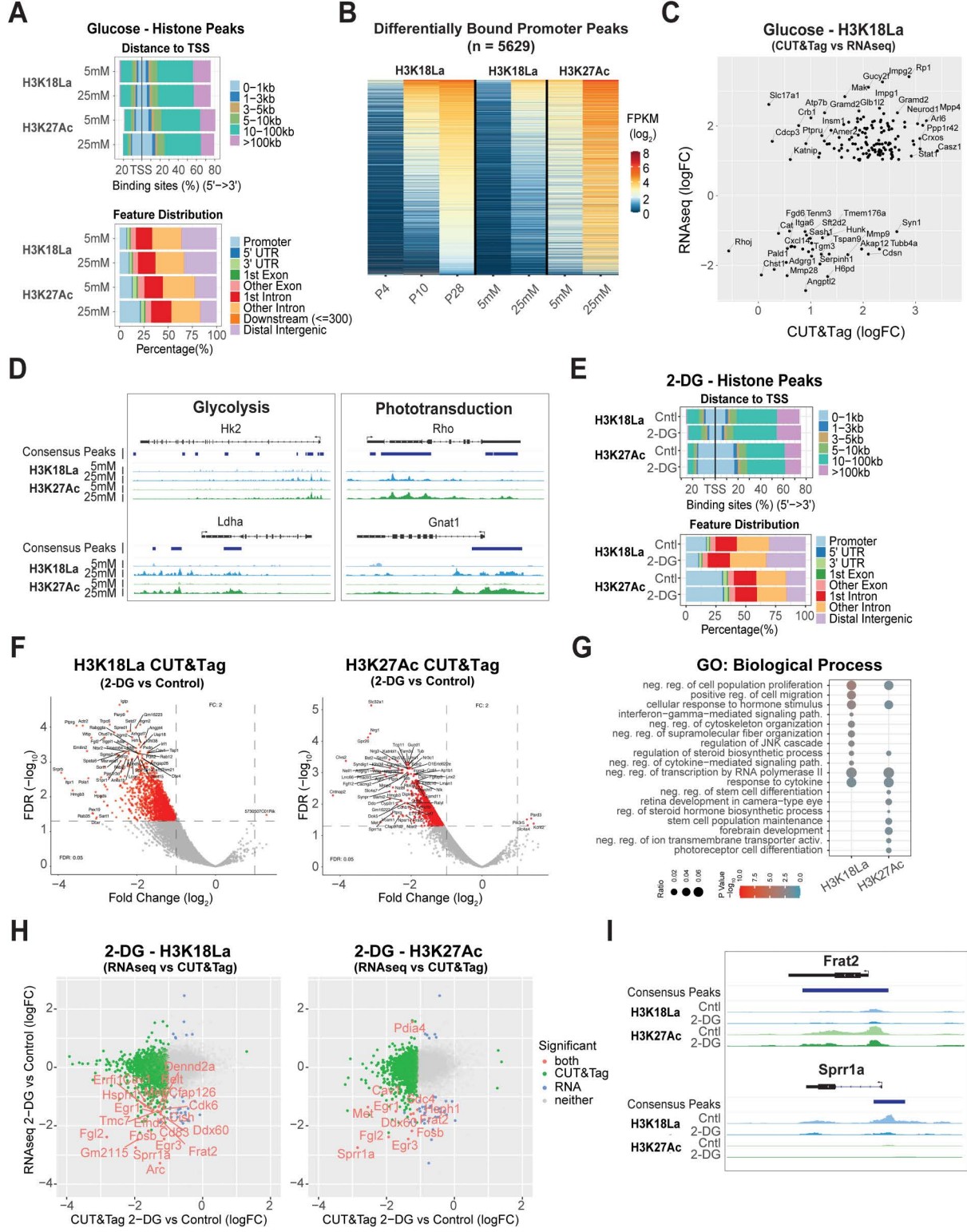

**Fig 6. Genome-wide profiling of H3K18La in retinal explant cultures with different glucose concentration.** (A) Upper panel: Percentage of H3K18La and H3K27Ac bound consensus peak loci distance relative to TSS for 5 mM and 25 mM glucose conditions (48 hours). Lower panel: Percentage of H3K18La bound consensus peak loci to gene annotation feature for 5 mM and 25 mM glucose conditions (48 hours). (B) Heatmap of peaks found from time course H3K18La differential binding analysis (Fig 5B) plotted with peak quantitation from explant CUT&Tag analysis of H3K18La and

H3K27Ac. (C) Comparison of significantly, differentially expressed genes from RNA-seq explant analysis (25 mM versus 5 mM) with explant CUT&Tag differential binding results. (D) Genomic histogram traces of explant histone marks for representative genes of several affected pathways observed. The histogram traces are group normalized for each gene. The bars under each gene track represent consensus peaks. (E) Upper panel: Percentage of H3K18La and H3K27Ac bound consensus peak loci distance relative to TSS under control (no inhibitor) and 2-DG (20 mM, 48 hour) conditions. Lower panel: Percentage of H3K18La bound consensus peak loci to gene annotation feature. (F) Left and Right panel: Differential binding analysis (2-DG versus control) for H3K18La (Left) and H3K27Ac (Right). Horizontal line: FDR = 0.05; vertical lines: |log2FC| = 1 (fold-change = 2). Red dots are significant differentially bound peaks. (G) Gene Ontology (Biological Process) enrichment for genes linked to differentially bound peaks for H3K18La and H3K27Ac. (H) RNA-seq differential expression versus CUT&Tag differential peak binding results for 2-DG versus control. Labeled red dots indicate genes significantly changed in both experiments. Green dots indicate genes significant in CUT&Tag only, blue dots for genes in RNA-seq only, and grey dots for genes significant in neither experiment. (I) Genomic histogram traces of explant histone marks for representative genes of several affected pathways observed. The histogram traces are group normalized for each gene. The bars under each gene track represent consensus peaks. Abbreviations: TSS, transcription start site; TES, transcription end site; logFC, log fold-change; CPM, counts per million; DB, differential bound; FPKM, fragments per kilobase per million reads; logFC, log fold-change.

resulted in broad reduction of H3K18La occupancy compared to controls (Figs 6F, Left and S4D and S11 Table), whereas H3K27Ac promoter peaks were moderately affected (Figs 6F, Right and S4D and S12 Table). The H3K18La peaks that show a decline by 20 mM 2-DG treatment were at genes enriched in immune pathways, consistent with a previous report [7], whereas a decline in H3K27Ac peaks was associated with photoreceptor and retinal development terms (Fig 6G and S13 Table). A majority of the differentially bound peaks were acquired from P4 to P28 (S4E Fig). Our transcriptomic data revealed a modest change in expression profile, with a trend of global repression (S4F Fig and S14 Table). Gene ontology analysis of differentially expressed genes after 2-DG treatment indicated selective activation of pathways related to ER stress and protein folding, whereas interferon response and immune activation gene networks were suppressed (S4F Fig, lower panel and S15 Table). We also detected a modest but noticeable relationship between changes in promoter H3K18La occupancy and transcriptional output (Fig 6H, Left). A subset of genes, including *Sprr1a* (signals neuronal injury and stress response), *Frat2* (regulates Wnt/β-catenin signaling), *Egr3* (transcription factor associated with neuronal activity) and *Fgl2* (immunomodulatory protein), showed coordinated downregulation at both the chromatin and RNA levels, suggesting that loss of glycolytic flux reduces lactylation at promoter regions and consequently their expression. However, a majority of loci exhibited weaker coupling, indicating that epigenomic changes may precede or only partially translate into transcriptional outcomes under 2-DG treatment. In contrast, H3K27Ac showed a lower set of correlated loci, largely overlapping with those observed in lactylation with concordant reduction in both acetylation and RNA levels; the exception being that of *Pdia4* (associated with ER-stress response), which showed higher expression (Fig 6H, Right). Select examples of genes including *Sprr1a*, *Frat2*, *Egr1* and *Pdia4* demonstrated the dynamics of H3K18La and H3K27Ac binding respectively in 2-DG versus control retinas (Figs 6I and S4G). A limitation of our 2-DG experiments was the lack of a concurrent cell death analysis. The inhibition of glycolysis is known to be cytotoxic, and therefore, we cannot definitively disentangle the epigenetic effects of metabolic substrate depletion from the broader, non-specific consequences of cellular stress and the initiation of apoptosis. The use of a non-metabolic, cytotoxic agent would be required in future studies to determine the specificity of the observed lactylation changes.

Taken together, these results demonstrated an association between glucose availability, H3K18La landscapes, and transcriptional response in retinal explants with partially overlapping yet distinct expression outcomes. Notably, many of these same genes also show changes in H3K27Ac binding in retinal explants with different conditions, suggesting that lactylation cooperates with canonical acetylation to reinforce transcriptional activity under changing metabolic conditions.

### Accessible chromatin regions with H3K18La are enriched for GC rich motifs

To explore how H3K18La impacts gene expression, we evaluated accessible chromatin regions containing H3K18La for TF binding motifs (S5A Fig). Comparison of TF motifs in accessible footprints at enhancers, promoters and gene bodies with or without the H3K18La peaks detected the enrichment of hundreds of TF binding motifs at H3K18La loci (Fig 7A). Enhancers

and promoters with H3K18La in accessible regions presented only 100–200 enriched motifs from different TFs compared to an array of TFs with enriched motifs in gene bodies (Fig 7A and 7B), highlighting their distinct and specific regulatory roles. Notably, enriched TF motifs, especially in promoters and enhancers, exhibited a clear bias toward GC-rich sequences (Figs 7C and S5B-S5D), consistent with a previous report [37]. TF motifs enriched at H3K18La promoters and enhancers included several proteins of the KLF and the SP families as well as KMT2A, ZBTB14, TFDP1, VEZF1, PATZ1 and CTCF (Figs 7C, S5 and S6 and S16–S24 Tables). Interestingly, top motifs enriched at H3K18La gene bodies were for different TFs, such as SREBF2, MAZ, ZNF425, SALL4, HAND1 and 2 and SMAD4 (S6 Fig and S16-S24 Tables). At specific loci, e.g., Got2 and Eno1, the accessible motifs at enhancers or promoters were concentrated within a single accessible foot-print of each H3K18La peak (Fig 7D-7E), whereas accessible motifs in the gene body of Glis1 gene were distributed among several accessible footprints (Fig 7F). In retinal explants, a similar preference for accessible GC rich motifs is evident at H3K18La loci (S5F-S5G Fig). In retinal explants, similar to P28 retina, accessible H3K18La regions within gene bodies showed a variety of motifs compared to enhancers or promoters (Fig 7G and 7H). Additional motifs are enriched in 25 mM compared to 5 mM glucose conditions for H3K18La, but not for H3K27Ac loci (Fig 7G and 7H).

We conclude that H3K18La loci are enriched for GC rich accessible footprints. Presence of different motifs in enhancer/promoter versus gene body points to specific functions of H3K18La at distinct genomic locations.

## Discussion

Development, homeostasis and response to environment are orchestrated by cooperative and coordinated actions of genetic factors and epigenome modifications, which together facilitate precise patterns of gene expression [39]. Cellular metabolism, and consequently availability of specific metabolites, can influence genome architecture [40]. Notably, histone modifications play critical roles in establishing a permissive chromatin state for activating or repressing transcription [6]. Unique physiological requirements of terminally differentiated retinal neurons present an attractive paradigm to explore the metabolism-epigenome nexus in development, aging and disease [39–41]. In this report, we explore the direct role of aerobic glycolysis and lactate levels in modulating histone H3K18La and its contribution to gene expression. We show that enhanced lactate levels during retinal development and under high glucose conditions lead to changes in genome-wide profiles of H3K18La. The correlations of H3K18La to H3K27Ac and H3K4me3 profiles and to gene expression demonstrate a key role of histone lactylation in fine-tuning expression of both tissue-specific and ubiquitous genes.

Translation of genomic information is influenced by reversible epigenetic modifications of histones, which can help balance the transcriptional output under distinct cellular environments. The discovery of acyl marks on histones, such as propionyl, butyryl, crotonyl, and the more recent lactyl-lysine, has greatly enhanced our knowledge of functional diversity of histone modifications [5]. Histone acylations can mark genomic regions of embryonic tissues to generate distinct cell types [14] and their profile provide a snapshot of the metabolic status of a given cell type [37,42]. Although many acylations including histone acetylation are established by overlapping acyltransferases, the local nuclear acyl-CoA pools can provide the specificity layer to their recruitment at specific loci [43].

Our data support a model in which lysine lactylation, particularly H3K18La links lactate metabolism to transcriptional programs driving retinal neuron maturation. Together with recent reports that alanyl-tRNA synthetase 1 (AARS1) can catalyze nuclear protein lactylation, these findings underscore the need to define how nuclear lactyl-CoA availability gates lactylation reactions and shapes the balance among competing marks [44]. Another key mechanistic question from our findings is what determines substrate choice for promiscuous acyltransferases like p300, which catalyze both lactylation and acetylation; therefore, future studies are needed to determine whether the changes in H3K18La we report are driven by lactyl-CoA availability in the nucleus or by direct competition with acetyl-CoA at the catalytic site. We show that unlike the broad activation mark H3K27Ac, H3K18La appears to involve recruitment of select TFs. Motif analysis reveals an enrichment of GC-rich accessible sequences at H3K18La peaks that include motifs associated with binding by zinc finger proteins and members of the SP and KLF families. The specificity of TF recruitment may be shaped both by relative TF

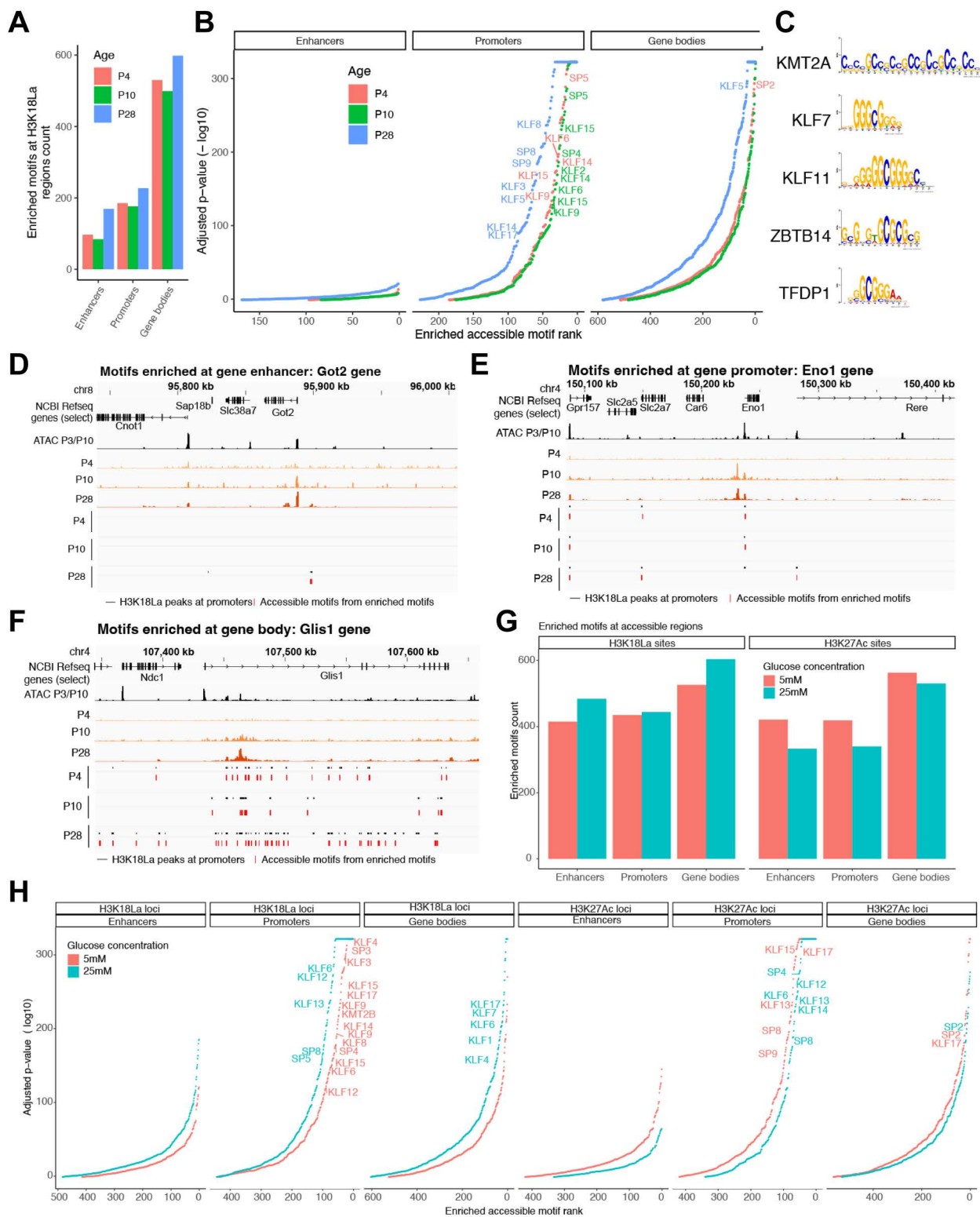

**Fig 7. H3K18La promoters are enriched for GC rich binding motifs.** (A) Number of enriched accessible motifs in H3K18La enhancers, promoters and gene bodies, at P4, P10 and P28. (B) Motif rank vs. p-value for motifs found at enhancers, promoters or gene bodies. (C) Top 5 accessible motifs enriched at H3K18La promoters at P28. (D-F) Examples of genes with H3K18La promoters containing accessible enriched motifs. Tracks represent

H3K18La signal for P4, P10 and P28, H3K18La peaks (black) at either enhancers (D), promoters (E) or gene bodies (F) and accessible footprints containing enriched motifs at the given peaks (red) at P4, P10 and P28, and a selection of Refseq genes. One accessible footprint can contain multiple enriched motifs. (G) Number of enriched accessible motifs in H3K18La or H3K27Ac enhancers, promoters and gene bodies, for 5- or 25-mM glucose in explants. (H) Motif rank vs. p-value for motifs found at H3K18La or H3K27Ac enhancers, promoters or gene bodies for 5- or 25-mM glucose in explants.

dosage in the retina and by competition among factors with highly similar motifs. However, more direct metabolic labelling experiments, combined with histone proteomics, are needed to quantify incorporation into lactyl-lysines and to test how perturbing nuclear lactyl-CoA pools shifts TF recruitment.

Time periods between P6 and P10 and before eye opening (at around P14) mark significant developmental milestones in mouse vision, with major changes in photoreceptor morphogenesis [45] and visual cortex [46]. Our results showing changes in energy homeostasis during these developmental periods are consistent with retinal morphogenesis and specific upregulation of metabolic, phototransduction and developmental genes [38]. The observed augmentation of H3K18La at specific loci during this stage suggests that histone lactylation may act as a metabolic sensor, linking increased glycolytic activity to transcriptional programs required for retinal maturation. Though precise mechanisms remain unclear, our findings support a model in which glycolysis/lactate/H3K18La regulatory axis provides a means of fine-tuning gene expression in response to developmental and/or metabolic demands. Interestingly, parallels can be drawn from other systems, such as Glis1-driven reprogramming, where a metabolite-epigenome feedback loop enhances glycolysis and lactate levels to modulate H3K18La and transcription of pluripotency genes [47].

Extending these observations, the high glucose condition in our explant retina cultures results in augmented H3K18La marks around the TSS of several photoreceptor genes (such as *Impg1, Impg2, Rp1, Gucy2f*). Although our CUT&Tag analysis reveals changes in H3K18La and H3K27Ac peaks upon glycolytic inhibition by 2-DG, these alterations are not paralleled by robust gene expression changes within the 48-hour timeframe used for generating our RNA-seq dataset. One possible explanation is that epigenetic remodeling precedes transcriptional changes and functions as an early priming event rather than producing immediate transcriptional changes. Thus, a more extended time-course analysis may be necessary to capture the full transcriptional impact of these chromatin changes. Also, 2-DG may have broader metabolic effects that likely contribute to the observed gene expression patterns. Moreover, our analysis focused on H3K27Ac as a control marker for transcriptionally active chromatin in response to metabolic shifts, though we acknowledge that a direct comparison with H3K18Ac would resolve the site-specific interplay between lactylation and acetylation at this residue under such conditions. Further, future studies combining high glucose with glycolysis inhibition or LDH inhibition (e.g., oxamate) could help discriminate whether the observed lactylation changes require glycolytic flux and/or lactate production.

Additional investigations are also needed to explore energy-independent functions of lactate and H3K18La-mediated transcriptional regulation in response to circadian rhythms, diet changes as well as during aging and disease. Changes in glucose availability and/or utilization with aging, as reported in rods [48], may have implications for wider epigenome modifications (including histone lactylation and acetylation) that make the retina vulnerable to disease. Our study thus presents a framework to elucidate the complex interplay among metabolism, gene regulation, and environmental factors and for understanding multifactorial etiology of age-related retinal diseases such as age-related macular degeneration (AMD) and diabetic retinopathy (DR) where changes in glycolytic flux are quite evident [49,50].

In conclusion, we show that H3K18La contributes to remodeling of the local chromatin environment and transcription in response to changes in metabolic state of the retina. Dynamics of histone lactylation at specific cis-regulatory elements likely fine-tunes gene expression patterns during retinal development and under distinct metabolic environments. The reversible nature of histone lactylation presents an opportunity for designing therapies as well as dietary paradigms as indicated by epidemiological studies [51,52].

### Limitations of the study

Our studies provide a direct link between aerobic glycolysis and gene expression via H3K18La-mediated changes in chromatin environment. However, we appreciate the limitations of the retinal explant cultures as well as OCR/ECAR measurements from ex vivo retina cultures, which may be limited by oxygen supply. Additional time course experimentation and different metabolic drugs may be helpful to complement our retinal explant experiments. We also acknowledge that our high-glucose retinal explant model, while physiologically relevant, does not exclude contributions from other metabolic shifts such as an altered cellular redox state or ATP levels to observed changes in histone lactylation. Our CUT&Tag studies provide genomic occupancy of H3K18La in the whole retina. As technical methods advance, single cell studies may uncover a more in-depth functional understanding of H3K18La in retinal cell types and its dysregulation in aging and disease.

## Materials and methods

### Ethics statement

All procedures involving mice were approved by the National Eye Institute Animal Care and Use Committee. The approval number is NEI-ASP# 650.

### Mouse models

C57BL/6J mice expressing EGFP under the control of the *Nrl* promoter (Nrlp-EGFP mice) [53] were used to perform Seahorse assays in the retina. Wild type mice (C57BL/6J) obtained from Jackson Laboratory were used for all other experiments unless specified. Mice were kept in a 12 light/12 dark hour cycle and fed *ad libitum* at the NEI animal facility.

### Retina Isolation and Explant Culture

Wild-type, P28 C57BL/6J mice used for retinal explant cultures were euthanized by $CO_2$ asphyxiation and freshly dissected in 1X Hank's Balanced Salt Solution (HBSS). To avoid variability due to circadian effects, retinas were dissected 2 hours after lights are turned ON in the facility for all experiments. Extraocular tissue was trimmed off, and the cornea and iris were carefully removed. Sclera along with the RPE was gently removed to get clean retina tissue. The dissected retinas were incubated in Neurobasal-A media supplemented with 5 mM glucose/25 mM glucose or 5 mM glucose with different concentration of 2-DG (10 mM and 20 mM for immunoblots and 20 mM for CUT&Tag, range that is typically used in cell culture studies), 0.2% B27 supplement, 0.1% N2 supplement, 0.1% Glutamax and 1X penicillin/streptomycin (Thermofisher Scientific) with 5% CO2 at 37 °C for 48 hours as described with daily medium replacement of 50% media with pre-warmed conditioned medium [18]. At the end of incubation period, the retinas were rinsed with prewarmed Krebs' Ringers medium and were further incubated in 0.3 mL Krebs' Ringers medium for 1 hour in 37 °C incubator. The supernatant and retinas were rapidly frozen separately at the end of the experiment for lactate assay and western blot respectively. For CUT&Tag and RNAseq, the retinas were processed immediately.

### Histone extraction

Nuclear fractions were prepared from flash-frozen whole retina tissues obtained from four biological replicate of C57BL/6J mice samples of different age groups (P2, P6, P10, P14 and P28). In brief, 50–100 mg of tissue was Dounce homogenized on ice in hypotonic lysis buffer (10 mM Tris-HCl, 10 mM NaCl, 3 mM $MgCl_2$, and 0.1% NP-40 alternative) with histone deacetylase and protease inhibitors and 1X phosphatase inhibitor. Crude sub-cellular fractions were separated by differential centrifugation at 800 x g for 10 minutes at 4 °C. The crude nuclear pellet was then washed twice with ice-cold 1X PBS prior to acid extraction of histones using 0.4 N $H_2SO_4$. The histones proteins were then precipitated using trichloroacetic acid with 0.1% sodium deoxycholate, washed with 100% acetone followed by elution in ultrapure water.

## Label-free chemical derivatization and mass spectrometry

Histone extracts from adult retina (n = 3, P28) tissue samples were subjected to propionic anhydride chemical derivatization to react primary amines, as described [54]. This reaction prevents derivatized lysines to be substrates for trypsin, thus restricting proteolysis to arginine sites, and reduces the overall charge state of the corresponding peptides making them amenable to LC/MS analyses. After derivatization, proteins were digested with sequencing grade modified porcine trypsin (Promega) overnight at 37 ºC. Tryptic peptides were further treated with propionic anhydride to label newly generated N-termini. Derivatized peptides were desalted by reversed phase solid phase extraction using Pierce C18 Spin columns. Peptides were loaded on a 30 cm x 75 µm ID column packed with ReproSil 1.9 µm C18 particles (Dr. Maischt) and resolved on a 5–35% acetonitrile gradient in water (0.1% formic acid) using a Thermo Scientific Easy nLC 1200. Eluting peptides were analyzed by an Orbitrap Fusion Lumos mass spectrometer (Thermo). The MS was set to collect 120,000 resolution precursor scans (m/z 380–2000 Th). Precursor ions were selected for HCD fragmentation at stepped 28,33,38% NCE in a data-dependent manner and spectra were collected in the orbitrap at 60,000 resolutions with first mass locked to 100 Th. In our mass spectrometry search parameters, lysine lactylation was identified as a variable modification with a mass shift of +72.021 Da, which is annotated by the search engine as 'Carboxyethyl' (S1 Table). For clarity and consistency with established biological nomenclature, all 'Carboxyethyl' annotations are referred to as 'Lactylation' (KLa) throughout the manuscript and figures.

## Immunohistochemistry

Eyes from 6-, 10-, 14- and 28-day-old C57BL/6J mice were enucleated, and the whole eyes were fixed in 4% (w/v) paraformaldehyde in phosphate-buffered saline (PBS) for 1 hour at room temperature. After being washed in PBS, the eyes were cryoprotected by sequential incubation in 10% and 20% sucrose in PBS for 1 hour at room temperature, followed by overnight incubation in 30% sucrose in PBS at 4 °C. After the cornea, lens, and vitreous body were removed, the eyecups were embedded in optimal cutting temperature (OCT) medium, cut vertically at 10 µm on a Leica CM1850 cryostat (Wetzlar, Germany), and stored at -80 °C until further use. Retinal sections were washed in PBS and blocked in 5% normal donkey serum in 0.5% Triton X-100 dissolved in filtered PBS for 1 hour at room temperature. Tissue sections were then incubated overnight at 4 °C with the following primary antibodies: rabbit monoclonal anti-H3K18La (1:100, PTM1406RM, PTM Bio, China), or rabbit monoclonal anti-L-lactyllysine (Pan-lactyl) (1:200, PTM-1401RM, PTM Bio, China), mouse monoclonal anti-H3K27Ac (1:100, ThermoFisher Scientific Invitrogen 2D7B3), mouse monoclonal anti-H3K27me3 (1:100, Active motif, 61018) or human anti- NRL (1:100 R&D Systems, AF2945). Following three washes in PBS, slides were incubated with Alexa Fluor 488 donkey anti-rabbit IgG (H + L) (1:5000, ThermoFisher Scientific Invitrogen, A-21206), Alexa Fluor 488 donkey anti-goat IgG (1:5000, ThermoFisher Scientific Invitogen A32814TR), Alexa Fluor 568 donkey anti-mouse IgG (1:5000, ThermoFisher Scientific Invitogen A10037) and 1 µg/ml of 4',6-diamidino-2-phenylindole (DAPI) (Sigma-Aldrich, D8417-1MG) for 1 hour at room temperature. Sections were then washed three times in PBS and mounted using Fluoromount-G mounting medium (SouthernBiotech, Birmingham, AL). Images were acquired with a Leica TCS SP8 at 40X or 63X magnification in confocal or lightning mode. The colocalization analysis on the images was performed using the Coloc module in Imaris v11.00 (Oxford Instruments). Costes randomization method was used to threshold the images and segmented nuclear areas were defined as regions of interests (ROI) to calculate voxel based colocalization.

## Seahorse assay

Extracellular acidification rate (ECAR, as a measure of glycolysis flux) and Oxygen consumption rate (OCR, as a measure of mitochondrial respiration rate) were assessed on different age groups of Nrl-GFP mice (P2, P4, P6, P10, P14 and P28) using Seahorse XFe24 Analyzer (Agilent, Santa Clara, CA, United States). Freshly dissected *ex vivo* retinal punches were prepared on the same day of the assay as described [55]. The assays were performed using Agilent Seahorse XFe24 Extracellular Flux Assay kits and Seahorse XF24 islet capture microplates, with the Seahorse DMEM medium (6 mM of

glucose, 0.12 mM of pyruvate, and 0.5 mM of glutamine). Four drugs: Oligomycin (final concentration 5 uM), Rotenone/ Antimycin A (final concentration 1 uM) and 2-DG (final concentration 50 mM) were used in the assay. Basal glycoECAR and basal mitoOCR were determined using the measurement at 36 minutes, right before addition of the first drug, with formulas described previously [55]. Glyco ATP production rate and mito ATP production rate were calculated using Agilent's ATP production rate calculation algorithm

(https://www.agilent.com/cs/library/whitepaper/public/whitepaper-quantify-atp-production-rate-cell-analysis-5991-9303en-agilent.pdf).

## Immunoblotting

Histone proteins were separated using 4–15% SDS-PAGE and transferred onto polyvinylidene fluoride (PVDF) membranes (Millipore). Blotted membranes were blocked in Licor blocking buffer and incubated with primary antibodies at 4 °C overnight anti-Pan lactyl (Rabbit, PTM1401, PTM Bio, China), anti-H3K18La (Rabbit, PTM1406RM, PTM Bio, China), anti- H4K12La (Rabbit, PTM101, PTM Bio, China), anti-H3K27Ac (Rabbit ab4729, Abcam, Cambridge, UK), anti-Histone H3 (Rabbit, ab176842, Abcam, Cambridge, UK). Membranes were then washed three times with TBS-T and incubated with secondary antibodies at room temperature for 1 hour. The immunoreactive products were detected using Licor Odyssey Imaging System. ImageStudio was used to perform quantitative analysis of western blot results as the ratio of the band intensities of target histone marks to the band intensities of reference proteins (H3). The specificity of anti-H3K18La antibody (PTM1406RM) was previously validated by peptide dot blot, confirming no cross-reactivity with H3K18Ac or H3K27Ac [7].

## Lactate assay

Lactate assay was performed on retinal tissue samples from different age groups of mice (P2, P6, P10, P14 and P28) as well as on the conditioned medium in the retinal explants using Promega Lactate-Glo (Promega J5021) assay according to the manufacturer's instructions. Retina samples was homogenized in 1 ml homogenization buffer with Inactivation solution at an 8:1 ratio using mechanical homogenizer. Tissue homogenate was used for protein estimation (Pierce BCA, 23225). Tissue lysates were neutralized with 0.125 ml neutralization Solution and incubated with 50 µl detection reagent. For explants, 50 µl conditioned media was incubated with 50 µl detection reagent. Luminescence was recorded after 1 hour incubation using Promega GloMax Plate Reader. The luminescence signal is further normalized with total protein concentration. The normalized luminescence values were plotted as proportional to lactate in the sample.

## CUT&Tag assay

CUT&Tag protocol was performed using CUT&Tag-IT Assay Kit – Tissue (Active Motif, Cat# 53171) with slight modifications. In brief, retinas were dissected at three timepoints (P4, P10 and P28) and cells were dissociated in a 5 ml polypropylene round-bottom tube with a papain dissociation protocol adapted from previous study [56]. After dissociation the cells were washed using 1X wash buffer provided in the kit at 200 x g for 5 minutes. CUT&Tag was performed using 500,000 cells per antibody following kit protocol. Antibodies against H3K18La (Rabbit, PTM1406RM, PTM Bio, China), and H3K27Ac (Rabbit ab4729, Abcam, Cambridge, UK) were used at a concentration of 1:100 in 100 µl and protein A-Tn5 (pA-Tn5) transposase (generous gift of Dr. Steven Henikoff, Howard Hughes Medical Institute, Washington, USA) used at a concentration of 700 ng/ml. Libraries were paired-end sequenced to 101 bases using the NextSeq 2000 platform (Illumina, San Diego, CA).

## RNA extraction and library preparation

Total RNA from explant retinae was extracted using TRIzol (Invitrogen, Carlsbad, CA), treated with DNase and cleaned up using the MagMAX mirVana Total RNA Isolation Kit (Applied Biosystems, Foster City, CA) following the manufacturer's instructions. Libraries were constructed with SMARTer Stranded Total RNA-Seq Kit v2 – Pico Input Mammalian (Takara Bio USA, Mountain

View, CA) with 4 ng of RNA and 13 PCR cycles library amplification. Paired-end sequencing of 101 bases was obtained using the NextSeq 2000 (Illumina, San Diego, CA). RNA-seq analysis pipeline was employed as previously described [38].

## CUT&Tag Analysis

H3K18La CUT&Tag fastq files were processed using nf-core/cutandrun v3.1 to GRCm38 assembly 10.5281/ zenodo.5653535 [57]. MACS peaks having a post facto qValue (-log10) > 6 and present in at least two biological replicates were used as consensus peaks for each experimental grouping. Peaks overlapping regions in the ENCODE mm10 blacklist (v2) were excluded during peak calling to minimize any false-positive enrichment. Further analysis was performed in R v4.2.3 (r-project.org). Annotation of consensus peaks was performed with ChipSeeker v1.34.1 [58] using Ensembl v102 annotation with promoter defined from -1000 bp to +500 bp from transcription start site (TSS). Enhancer regions are defined as being -10kb to -1kb from TSS. Gene enrichment analysis was performed using clusterProfiler v4.6.0 with Gene Ontology Biological Process database (2022-Sep12 release). Peak quantitation for differential binding analysis was performed on merged consensus peaks P4, P10, and P28 using CSAW v1.32.0 [59]. Normalization was performed using windowCounts function in CSAW with 10,000 base windows. PCA analysis was performed using PCAtools in R. Differential peak analysis was performed using gene wise negative binomial generalized linear models with quasi-likelihood F-tests using edgeR v3.40.2 [60]. Explant analysis PCA was performed using peak quantitation from merged H3K18La and H3K27Ac consensus peaks. Previously published retina data was used in comparative analysis of H3K27Ac and H3K4me3 ChIP-seq marks [3] and NRL CUT&RUN [61]. Previously published H3K18La peaks from different tissues [37] were processed using the same pipeline as the retina data for comparison. Upset plot for tissues was created from UpSetR v1.4.0 from presence of H3K18La peak in the promoter of protein coding genes. Hierarchical clustering (Ward's method) of the Euclidean distance matrix of shared H3K18La promoter peaks between tissues was generated from the phi coefficient for all pairwise comparisons.

## Motifs enrichments analysis

We identified enriched motifs at H3K18La overlapping promoters, enhancers or gene bodies at P4, P10, P28, explant 5 mM or explant 25 mM glucose over control data. For each age ~ region type pairs (e.g., P4 ~ promoters), we selected the control data by taking all regions for the same type not overlapping any H3K18La at this age (e.g., all promoters without H3K18La peak at P4). We called footprint regions on these lists of regions (peaks and controls) using rgt-hint foot printing [62] with the ATAC-Seq option, using mapped ATAC-seq reads (SRR5884808, SRR5884809, SRR5884805, SRR5884806) [3]. We extracted the DNA sequences from all footprint regions using bedtools getfasta [63]. Finally, we ran motif enrichment analysis using MEME tool AME [64] (v5.5.7), with default parameters on peaks vs. control footprints, using the motifs database HOCOMOCO v12 [65] for Human and Mouse.

## Statistical analysis

Statistical analyses and corresponding p-values for immunoblot and Seahorse experiments were performed in Graph-Pad Prism (v10.3.1), being reported within each plot or detailed in the respective Figure legends. A one-way ANOVA with post hoc Tukey's test was used for Immunoblots for Pan-lactyl, H3K18La, and H4K12La for different age groups of mice. A one-way ANOVA followed by Sidak's multiple comparisons test was used to compare the experimental versus control conditions in retinal explants. All data are presented as mean ± SEM. *, $p < 0.05$

## Supporting information

**S1 Fig. Mass spectrum H3.3 peptide and genome-wide profiling of H3K18La, related to Fig 2B and Fig 4, respectively.** (A) Annotated HCD Spectrum of Histone H3.3 peptide R.KQLATKAAR.K modified with 1 × Propionyl [N-Term]; 1 × Propionyl [K6]; 1 × lactylation [K]; 1 × Acrolein [K1]. Peaks in blue are fragment ions containing the C-terminus (y-ions) and peaks in red are fragment ions containing the N-terminus (b-ions). (B) Genomic histogram traces for 1 megabase

regions of H3K18La sample replicated at each timepoint for genes involved in glycolysis (*Aldoa* and *Ldha*) and the phototransduction cascade in rod photoreceptors (*Rho*). The histogram traces are group scaled for each individual timepoint. The bars under each timepoint histogram represent consensus peaks. (C) GO Biological Process gene sets enriched for genes containing H3K18La in each timepoint in different defined gene regions.
(TIF)

**S2 Fig. H3K18La peak characterization and comparison, related to Fig 4.** (A) H3K18La peaks are overlapped with 16 chromatin states defined by ChromHMM from retina ChIP-seq data [3] for each defined gene region and timepoint. (B) Colocalization of H3K18La peaks with H3K27Ac [3] bound regions. (C) Confocal immunofluorescent images showing colocalization of NRL and H3K27me3 with H3K18La in nuclear periphery of photoceptor cells. Scale bars, 10 μm. (D) Genomic histogram traces of histone marks during development for representative selected genes found in Fig 4H. H3K18La marks at P4, P10, and P28 (Red), H3K27Ac [3] at P3, P10, P21 (Green), and H3K4me3 at P3, P10, and P21 (Purple). (E) Hierarchical clustering of shared H3K18La promoter peaks between retina and other tissues [37]. (F) GO Biological Process gene sets enriched for protein coding genes containing H3K18La in the promoter which are positively or negatively correlated with RNA-seq expression during development. Abbreviations: GAS, Gastrocnemius; MT, Post-mitotic end-state myotubes; PIM, Post- ischemia macrophages; MB, Myoblasts; ADIPO, Adipose tissues; BMDM, Bone marrow-derived macrophages.
(TIF)

**S3 Fig. Increase in glucose from 5 mM to 25 mM reshapes H3K18La/H3K27Ac landscapes in retinal explants, related to Fig 6.** The number of consensus peaks passing a 1x10-6 FDR for each replicate per explant glucose concentration and histone mark. The red dot indicates the number of genes containing a peak, whereas the blue dot represents the number of genes containing a peak in the proximal promoter. (A) Heatmap of H3K18La-bound and H3K27Ac-bound peak signal enrichment relative to distance from TSS/TES for all genes and their flanking 3 kb region. (B) Principal component analysis of quantitative peak binding for all H3K18La and H3K27Ac samples. (D) Genomic histogram traces for 1 megabase regions of H3K18La and H3K27Ac sample replicates at each glucose conditions (5 mM and 25 mM) for genes involved in glycolysis (*Ldha*) and the phototransduction cascade in rod photoreceptors (*Rho*). The histogram traces are group scaled for each individual timepoint. The bars under each timepoint histogram represent consensus peaks. (E) Genomic histogram traces of H3K18La and H3K27Ac in explants for representative genes of several affected pathways observed. The histogram traces are group normalized for each gene. The bars under each gene track represent consensus peaks. Abbreviations: FDR, false discovery rate; PCA, principal component analysis; PC1, principal component 1; PC2, principal component 2; DB, Differentially bound.
(TIF)

**S4 Fig. Glycolysis inhibition with 2-DG reshapes H3K18La/H3K27Ac landscapes in retinal explants, related to Fig 6.** (A) The number of consensus peaks passing a $1 \times 10^{-6}$ FDR for each replicate per explant 2-DG concentration and histone marks. The red dot indicates the number of genes containing a peak, whereas the blue dot represents the number of genes containing a peak in the proximal promoter. (B) Heatmap of H3K18La and H3K27Ac-bound peak signal enrichment for all genes and their flanking 3kb region under control (no inhibitor) and 2-DG (20 mM, 48 hour) conditions. (C) PCA of CUT&Tag quantitative peak binding separates controls from 20 mM 2-DG-treated retinal explants for H3K18La and H3K27Ac. (D) Differentially bound promoter peaks for H3K18La (Left) and H3K27Ac (Right) for 2-DG versus control in retinal explants. n denotes the number of peaks. (E) Number of genes containing differential bound promoter peaks for H3K18La from 2-DG versus control in retinal explants compared to retinal development (P4-P28). (F) RNA-seq expression analysis for retinal explants (2-DG versus contro). Horizontal line: FDR = 0.05; vertical lines: |log2FC| = 1 (foldchange = 2). Red dots are significant differentially expressed genes. Gene Ontology Biological Process analysis of genes differentially expressed in RNA-seq following 2-DG treatment versus control. (G) Genomic histogram traces of explant histone marks for representative genes of several affected pathways observed. The histogram traces are group normalized

for each gene. The bars under each gene track represent consensus peaks. Abbreviations: TSS, transcription start site; TES, transcription end site; PCA, principal component analysis; PC1, principal component 1; PC2, principal component 2; FDR, false discovery rate; logFC, log fold-change.
(TIF)

**S5 Fig. Enriched motifs during development for peaks containing H3K18La, related to Fig 7.** (A) Accessible motifs enrichment analysis pipeline. (B) Top 5 accessible motifs enriched at H3K18La promoters at P4 (left) and P10 (right). (C-D) Top 5 accessible motifs enriched at H3K18La enhancers (C) and gene bodies (D) at P4 (left) and P10 (center) and P28 (right). (E) Expression levels of expressed TF from the 100 top enhancers, promoters or gene bodies motifs at P4 and P10. (F-G) Top 5 accessible motifs enriched at H3K18La (F) or H3K27Ac (G) promoters, enhancers and gene bodies.
(TIF)

**S6 Fig. Enriched motifs in defined gene regions containing H3K18La, related to Fig 7.** Expression levels of expressed TF from the 100 top enhancers, promoters or gene bodies motifs at P28.
(TIF)

**S1 Table. KLa peptide mass spectrometry results, related to Fig 2B.** Results from mass-spectrometry analysis, the identified lactylation sites on histone proteins when searched through the entire mouse database at all the different histone isoforms. Lysine lactylation was set as a variable modification with a mass shift of +72.021 Da, which is annotated by the search engine as "Carboxyethyl" in the "Modification" column.
(XLSX)

**S2-S4 Tables. FGE results from H3K18La binding in promoter, enhancer, and gene body, related to S1C Fig.** Results from functional gene enrichment (FGE) analysis of Gene Ontology Biological Process for genes with H3K18La peaks located in the promoter (S2 Table), enhancer (S3 Table), and gene body (S4 Table) for each age investigated. GeneRatio indicates the number of genes identified in the pathway divided by the total number of genes at the age and peak location interrogated. BgRatio represents the number of genes in the ontology term divided by the total number of genes in the dataset. The 'p.adjust' column indicates the Benjamini-Hochberg false discovery rate.
(ZIP)

**S5 Table. FGE results from dynamic active histone mark (H3K4me3 and H3K27Ac) co-occupancy with H3K18La, related to Fig 3H.** Results from functional gene enrichment (FGE) analysis of Gene Ontology Biological Process for protein coding genes with H3K4me3, H3K27Ac, and H3K18La peak co- occupancy located in the promoter regions. GeneRatio indicates the number of genes identified in the pathway divided by the total number of genes at the age and peak location. interrogated. BgRatio represents the number of genes in the ontology term divided by the total number of genes in the dataset. The 'p.adjust' column indicates the Benjamini- Hochberg false discovery rate.
(XLSX)

**S6 Table. FGE results from H3K18La peaks unique to retina, related to Fig 3J.** Results from functional gene enrichment (FGE) analysis of Gene Ontology Biological Process for protein coding genes with H3K18La peaks in promoter regions unique to retina. GeneRatio indicates the number of genes identified in the pathway divided by the total number of genes at the age and peak location interrogated. BgRatio represents the number of genes in the ontology term divided by the total number of genes in the dataset. The 'p.adjust'column indicates the Benjamini-Hochberg false discovery rate.
(XLSX)

**S7 Table. Differential binding analysis results, related to Fig 4B.** Results from differential binding analysis of H3K18La peaks at P10 versus P4 and P28 versus P4. Results fit a quasi-likelihood (QL) negative binomial generalized log-linear model (GLM) from the normalized count data. Log2 fold-change results are indicated in logFC.P10vsP4 and logFC.P2P4

columns. False discovery rate (FDR) indicates the Benjamini-Hochberg false discovery rate of the GLM QL fit. Columns P4_FPKM_mean, P10_FPKM_mean, and P28_FPKM_mean are the mean peak quantitation values in fragments per kilobase of peak per million (FPKM) mapped reads.
(XLSX)

**S8 Table. FGE results from H3K18La peak quantitation correlated with RNA-seq expression, related to S2F Fig.** Results from functional gene enrichment (FGE) analysis of Gene Ontology Biological Process for quantitation of protein coding genes with H3K18La peaks in promoter regions during development which are positively and negatively correlated with RNA-seq expression from retina. GeneRatio indicates the number of genes identified in the pathway divided by the total number of genes at the age and peak location interrogated. BgRatio represents the number of genes in the ontology term divided by the total number of genes in the dataset. The 'p.adjust' column indicates the Benjamini-Hochberg false discovery rate.
(XLSX)

**S9 Table. FGE results from dynamic active histone mark (H3K4me3 and H3K27Ac) co-occupancy with H3K18La with positive correlation to RNA-seq, related to Fig 4E.** Results from functional gene enrichment (FGE) analysis of Gene Ontology Biological Process for protein coding genes with H3K4me3, H3K27Ac, and H3K18La peak co- occupancy located in the promoter regions. Dynamic occupancy is performed with genes showing positive correlation with RNA-seq gene expression during development. GeneRatio indicates the number of genes identified in the pathway divided by the total number of genes at the age and peak location interrogated. BgRatio represents the number of genes in the ontology term divided by the total number of genes in the dataset. The 'p.adjust' column indicates the Benjamini-Hochberg false discovery rate.
(XLSX)

**S10 Table. RNA-seq differential expression analysis results related to Fig 6C.** Results from differential expression analysis of genes from retinal explants supplemented with 25 mM versus 5 mM glucose. Results fit a quasi-likelihood (QL) negative binomial generalized log-linear model (GLM) from the normalized count data. $Log_2$ fold-change results are indicated in logFC column. False discovery rate (FDR) indicates the Benjamini-Hochberg false discovery rate of the GLM QL fit. Columns 5 mM_CPM_mean and 25 mM_CPM_mean are the mean gene expression values in gene counts per million (CPM) mapped reads.
(XLSX)

**S11-S13 Tables. Differential binding analysis and FGE results from 2-DG CUT&Tag of H3K18La & H3K27Ac related to Fig 6F.** Results from differential binding of 2-DG versus control for H3K18La (S11 Table) and H3K27Ac (S12 Table) CUT&Tag. Results fit a quasi-likelihood (QL) negative binomial generalized log-linear model (GLM) from the normalized count data. $Log_2$ fold-change results are indicated in logFC column. False discovery rate (FDR) indicates the Benjamini-Hochberg false discovery rate of the GLM QL fit. Columns H3K18LA_CNTRL_FPKM_mean and H3K18LA_2DG_FPKM_ mean are the mean peak quantitation values in fragments per kilobase of peak per million (FPKM) mapped reads. Results from functional gene enrichment (FGE) analysis of Gene Ontology Biological Process for the differential binding results (S13 Table). GeneRatio indicates the number of genes identified in the pathway divided by the total number of differentially bound genes interrogated. BgRatio represents the number of genes in the ontology term divided by the total number of genes in the dataset. The 'p.adjust' column indicates the Benjamini-Hochberg false discovery rate.
(ZIP)

**S14-S15 Tables. RNA-seq differential expression analysis and FGE results from 2-DG experiment related to S4F Fig.** Results from differential expression analysis of genes from retinal explants supplemented with 2-DG versus control (S14 Table). Results fit a quasi-likelihood (QL) negative binomial generalized log-linear model (GLM) from the

normalized count data. Log2 fold-change results are indicated in logFC column. False discovery rate (FDR) indicates the Benjamini-Hochberg false discovery rate of the GLM QL fit. Results from functional gene enrichment (FGE) analysis of Gene Ontology Biological Process for the differential expression analysis (S15 Table). GeneRatio indicates the number of genes identified in the pathway divided by the total number of differentially bound genes interrogated. BgRatio represents the number of genes in the ontology term divided by the total number of genes in the dataset. The 'p.adjust' column indicates the Benjamini-Hochberg false discovery rate.
(ZIP)

**S16-S24 Tables. TF motif analysis, related to S5 Fig.** List of TFs and their associated transcription measured by RNA-seq for P4 promoters (S16 Table), P4 enhancer (S17 Table), P4 gene body (S18 Table), P10 promoter (S19 Table), P10 enhancer (S20 Table), P10 gene body (S21 Table), P28 promoter (S22 Table), P28 enhancer (S23 Table), P28 gene body (S24 Table).
(ZIP)

## Acknowledgments

We thank members of Swaroop laboratory, especially Jayshree Advani, Zachary Batz, Anupam Mondal and Nivedita Singh, for discussions. This work utilized the computational resources of the NIH HPC Biowulf cluster (https://hpc.nih.gov).

## Author contributions

**Conceptualization:** Mohita Gaur, Claire Marchal, Anand Swaroop.

**Data curation:** Mohita Gaur, Matthew J. Brooks.

**Formal analysis:** Mohita Gaur, Matthew J. Brooks, Ke Jiang, Claire Marchal, Anand Swaroop.

**Funding acquisition:** Anand Swaroop.

**Methodology:** Mohita Gaur, Matthew J. Brooks, Xulong Liang, Ke Jiang, Anjani Kumari, Milton A. English, Paolo Cifani, Maria C. Panepinto, Jacob Nellissery, Robert N. Fariss, Laura Campello, Claire Marchal.

**Project administration:** Anand Swaroop.

**Resources:** Anand Swaroop.

**Software:** Matthew J. Brooks.

**Supervision:** Anand Swaroop.

**Validation:** Mohita Gaur, Matthew J. Brooks, Claire Marchal.

**Visualization:** Mohita Gaur, Matthew J. Brooks.

**Writing – original draft:** Mohita Gaur, Matthew J. Brooks, Claire Marchal, Anand Swaroop.

**Writing – review & editing:** Mohita Gaur, Xulong Liang, Ke Jiang, Anjani Kumari, Jacob Nellissery, Laura Campello, Claire Marchal, Anand Swaroop.

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
