## [Decision Letter · Decision Letter 0]

29 Jan 2026

PGENETICS-D-25-01094

Augmented lactate and histone H3K18 lactylation contribute to metabolic control of gene expression in the retina

PLOS Genetics

Dear Dr. Swaroop,

Thank you for submitting your manuscript to PLOS Genetics.

Your manuscript has been seen by 3 external reviewers; as you will see, they are generally positive but have a number of comments and concerns that we ask you address in a minor revision. I do not think any new experiments are needed but it will be important to moderate some of your conclusions and provide additional detail in response to the reviewers' comments.

Please submit your revised manuscript within by Feb 28 2026 11:59PM. If you will need more time than this to complete your revisions, please reply to this message or contact the journal office at plosgenetics@plos.org. Please include the following items when submitting your revised manuscript:

We look forward to receiving your revised manuscript.

Kind regards,

Gregory S. Barsh

Academic Editor

PLOS Genetics

John Greally

Section Editor

PLOS Genetics

Aimée Dudley

Editor-in-Chief

PLOS Genetics

Anne Goriely

Editor-in-Chief

PLOS Genetics

**Journal Requirements:**

At this stage, the following Authors/Authors require contributions: Anand Swaroop. Please ensure that the full contributions of each author are acknowledged in the "Add/Edit/Remove Authors" section of our submission form.

The list of CRediT author contributions may be found here: https://journals.plos.org/plosgenetics/s/authorship#loc-author-contributions

2) We noticed that you used the phrase 'data not shown' in the manuscript. We do not allow these references, as the PLOS data access policy requires that all data be either published with the manuscript or made available in a publicly accessible database. Please amend the supplementary material to include the referenced data or remove the references.

- ® on page: 29

- TM on page: 29.

Potential Copyright Issues:

i) Figures 1H, 2A, and 5A. Please confirm whether you drew the images / clip-art within the figure panels by hand. If you did not draw the images, please provide (a) a link to the source of the images or icons and their license / terms of use; or (b) written permission from the copyright holder to publish the images or icons under our CC BY 4.0 license. Alternatively, you may replace the images with open source alternatives. See these open source resources you may use to replace images / clip-art:

6) In the online submission form, you indicated that your data will be submitted to a repository upon acceptance. We strongly recommend all authors deposit their data before acceptance, as the process can be lengthy and hold up publication timelines. Please note that, though access restrictions are acceptable now, your entire minimal dataset will need to be made freely accessible if your manuscript is accepted for publication. This policy applies to all data except where public deposition would breach compliance with the protocol approved by your research ethics board. If you are unable to adhere to our open data policy, please kindly revise your statement to explain your reasoning and we will seek the editor's input on an exemption.

7) Please amend your detailed Financial Disclosure statement. This is published with the article. It must therefore be completed in full sentences and contain the exact wording you wish to be published.

2) If any authors received a salary from any of your funders, please state which authors and which funders..

8) Please send a completed 'Competing Interests' statement, including any COIs declared by your co-authors. If you have no competing interests to declare, please state "The authors have declared that no competing interests exist". Otherwise please declare all competing interests beginning with the statement "I have read the journal's policy and the authors of this manuscript have the following competing interests"

**Reviewers' comments:**

Reviewer's Responses to Questions

**Comments to the Authors:**

Reviewer #1: I have uploaded my comments as an attachment

Reviewer #2: This is an extensive analysis of histone lactoylation in mouse retinas. In carefully done and clearly reported studies the investigators show where histone lactoylation occurs and they show how it changes during development. They identify sites of lactoylation on histones and they quantify changes in gene expression that correlate with histone lactoylation. This report will be useful because it provides a foundation for future studies of this type of epigenetic modification and it identifies genes whose expression are likely to be affected by it.

The authors should address the following questions:

1. In Fig. 2D there is a lot of H2K18 lactoylation in the inner retina even at P6, but Fig. 1B shows that there is not very much lactate being produced at P6 (based on extracellular acidification). Please discuss or speculate on where the lactate could come from to be a substrate for a level of inner retina lactylation at P6 that appears to be equal to or to be even more abundant that the amount of lactoylation in the outer retina at P2.

2. I think it is not accurate to state: "Together, these results demonstrate that glycolytic inhibition by 2-DG perturbs both lactoylation and acetylation landscapes in retinal explants, with partially overlapping yet distinct expression outcomes, providing an additional layer of regulation at a subset of stress and immune response-related genes". I feel that this might be too strong an interpretation for the reasons that the authors clearly describe in the preceding paragraph. In the experiments in Fig. 7, 2-DG is used to block glycolysis. That likely makes the cells very stressed, i.e. ATP levels in the cell probably drop substantially. (In fact it would be worthwhile and probably not too difficult, to quantify ATP under these conditions.) The amount of lactoyl CoA substrate for histone lactoylation likely is diminished. Low ATP levels also must affect transcription overall so it would make the findings from these experiments difficult to interpret unambiguously. Very low levels of nucleotide triphosphates must slow transcription and it may do that differently for every gene being transcribed (probably influenced by relative rates of transcription) independently of any specific stress responses.

3. Many of the experiments in this report are done by raising the concentration of glucose to produce more lactate. I think the authors are making the assumption that the only, or the primary, relevant effect of increased glycolytic flux is that more lactate is produced. But there also could be other effects of increasing glucose, like more NADH, more pentose phosphate pathway activity, more NADPH or more ATP. Wouldn’t it be possible to address this by measuring effects on lactoylation and gene expression by directly adding lactate instead of glucose? Also, would addition of pyruvate cause the same effects as adding glucose or adding lactate? What patterns of lactoylation would be produced if glucose was added at a high concentration but a glycolysis inhibitor like iodoacetate or oxamate was added?

Reviewer #3: In this study, the authors focus on a potential connection between metabolism and gene expression via epigenetic control in the developing mouse retina. Specifically, they examine one histone modification, histone lactylation (H3K18-lactylation), that correlates with the increased glycolysis and lactate levels that they observe in the developing retina. This idea is intriguing because it would provide a really nice mechanism to connect metabolic changes with gene expression during retina development, and this could be particularly important for cells in the retina such as photoreceptors that are highly metabolically active. The authors provide some good correlative data supporting an increase in H3K18-lactylation that parallels glycolysis in the developing retina, but some of their conclusions are complicated by the fact that at least one acetylation mark – H3K27ac – seems to follow H3K18La very closely. This makes it difficult to separate out the impact of histone lactylation from acetylation. In addition, the transcriptional changes they observe don’t seem to correlate in one direction (positive or negative) with H3K18La, making it a bit hard to understand exactly what the role of this modification may be. Thus, some of the major conclusions of the paper as outlined in the title and abstract may be too strong based on the data provided. There are also some technical concerns regarding both the CUT&Tag analysis and antibody specificity that should be addressed because these data were critical for many of the conclusions of the study. Overall, this is an interesting study on an important area that could provide some insight into the connections between metabolism and the epigenome in the developing retina. We also want to commend the use of the retinal explant model that provided a very good approach to testing the role of glucose metabolism in altering chromatin marks and gene expression via direct manipulation of available glucose and 2-DG. Our specific major and minor concerns are outlined below.

Major concerns:

1. Apart from the mass spectrometry data in Fig. 2, most assays (WB, IHC, CUT&Tag) that assess H3K18La in this study used the same antibody: anti-H3K18La (Rabbit, PTM1406RM, PTM Bio). We recommend that the authors carefully check the specificity of this antibody to confirm that it doesn’t cross-react with H3K18ac or H3K27ac, especially when H3K18La and H3K27ac are found in many shared regions. Some discussion of this point should be included and/or specificity controls if appropriate.

2. Since H3K18La is deposited by histone acetyltransferase p300, which also acetylates H3K18ac and H3K27ac, it is possible that the observed H3K18La changes reflect a byproduct of p300-mediated acetylation activity. Therefore, inclusion of appropriate controls would be important to distinguish these possibilities. Specifically, in Fig.2D, in addition to pan-lactyl and H3K18La, we recommend assessing H3K18ac and/or H3K27ac, and pan-acetylation changes at least at two representative stages. In addition, if H3K27ac and H3K18ac turn out to change during development, acetyl-CoA levels should be measured as well.

3. In Fig.2B, the mass spectrometry data are important for supporting H3K18La as a relevant target, but the current analysis lacks quantification. The claim that “lysine lactylation is abundant across histones” is not supported by a figure. Moreover, it is unclear whether H3K18La is more abundant than other histone modifications and why it was selected for follow-up. Some quantification or discussion of the relative abundance of this particular mark relative to other histone marks (eg acetylation) would be helpful here.

4. We are a little concerned about the CUT&Tag data quality as no negative control (e.g., anti-IgG) was included. A negative control is important for reliable peak calling and for meaningful comparisons of signal across conditions since CUT&Tag, like other chromatin profiling techniques, often has some non-specific background. We therefore recommend that the authors either repeat at least one CUT&Tag experiment at P28 with the inclusion of an IgG control, or incorporate high-quality, stage-matched public IgG CUT&Tag data from mouse retina for all peak-based analyses. Considering the authors’ focus on the co-localization of H3K27ac and H3K18La, accounting for non-specific binding is necessary to reduce potential false-positives. In addition, we suggest including the ENCODE blacklist during peak calling. Based on the bigwigs shown for some of the P4 retinas, we are not sure that the peaks identified represent any real signal at this stage – background controls may help here as well.

5. In connecting the increase of H3K18La to transcriptomic data in Fig 4C, we are concerned that the analysis focuses exclusively on positively correlated genes, while almost an equal number of genes show negative correlations with increased H3K18La. In addition, the conclusion that H3K18La activates transcription in a permissive chromatin context appears to be based solely on the co-occurrence of H3K18La and increased gene expression at a small number of genes, which is insufficient to support a general activating role.

Minor concerns:

6. In Fig. 2C-D, the increase in H3K18La appears more pronounced in the ONL by IHC than by western blotting. Since IHC shows cell type-specific changes in H3K18La, whereas the WB samples represent mixed cell populations, we recommend quantifying IHC data rather than the WB results. Inclusion of some other histone marks (eg H3K27ac, H3K18ac) in these IHC data could also be helpful to show if this increase is unique to lactylation or not.

7. In Fig. 2C, only H3K18La increased with development whereas pan- and H4K12La do not significantly change. Considering the proposed model, it is unclear as to why H3K18La in particular is increasing. Is there a reason why this appears to be specific to H3K18La?

8. In Fig. 3C, the authors should check H3K18La levels on peaks rather than (or in addition to) on genes (tss) in the metaplots, which might provide more consistent results. Fig. 3B appears to be redundant. There is no y axis label in Fig. 3C.

9. Fig. 3E, to compare with colocalization of H3K18La and H3K27ac, we recommend first presenting the overall number of overlapping peaks or genes before highlighting region-specific overlaps.

10. The reproducibility of Fig.4’s P10 data for H3K18La CUT&Tag peaks is concerning. The authors address this by stating that the variability may be a result of active differentiation in these tissues. However, the RNA-seq data does not support the variability of replicates at the gene expression level. With only two replicates of great variability, the P10 H3K18La CUT&Tag is difficult to interpret.

11. In Fig. 5, what is the rationale for measuring H3K27ac levels rather than pan-acetyl or H3K18ac upon glucose or 2-DG treatment?

12. For the text relating to Fig. 6, stating that “Lactate promotes enrichment of H3K18La” is not entirely supported by the data. Lactate availability appears to correlate with H3K18La levels. However, only glucose was supplemented. Therefore, it cannot be determined whether lactate does indeed promote enrichment of H3K18La in these cells or another process impacted by glucose availability.

13. Fig. 6B appears to be unnecessary. In Fig. 6C and 6E, the increased H3K27ac signal in the metaplot seems to conflict with the conclusion in Fig. 5C that H3K27ac remains unchanged under high-glucose treatment. We suggest clarifying this apparent discrepancy.

14. Fig. 7 could be combined with Fig. 6 for better comparison. In Fig. 7C, the scale of the metaplot is not the same for control and 2-DG treatment. We would also like to see a metaplot for H3K27ac after 2-DG treatment. Similar to Fig. 6C, H3K18La levels show little change after 2-DG, which is inconsistent with the WB result in Fig. 5D.

15. Fig. 8 (motif enrichment) should be in the supplemental – this analysis doesn’t have a strong connection to the rest of the manuscript and there are no follow-up studies on the transcription factors identified.

**Have all data underlying the figures and results presented in the manuscript been provided?**

Large-scale datasets should be made available via a public repository as described in the *PLOS Genetics*
data availability policy, and numerical data that underlies graphs or summary statistics should be provided in spreadsheet form as supporting information., and numerical data that underlies graphs or summary statistics should be provided in spreadsheet form as supporting information., and numerical data that underlies graphs or summary statistics should be provided in spreadsheet form as supporting information., and numerical data that underlies graphs or summary statistics should be provided in spreadsheet form as supporting information.

Reviewer #1: Yes

Reviewer #2: Yes

Reviewer #3: Yes

PLOS authors have the option to publish the peer review history of their article (what does this mean?). If published, this will include your full peer review and any attached files.). If published, this will include your full peer review and any attached files.). If published, this will include your full peer review and any attached files.). If published, this will include your full peer review and any attached files.

...

Reviewer #1: **Yes:**Daniel HassDaniel HassDaniel HassDaniel Hass

Reviewer #2: **Yes:**James B. HurleyJames B. HurleyJames B. HurleyJames B. Hurley

Reviewer #3: No

**Figure resubmission:**
---

## [Editor Report · Decision Letter 1]

20 Mar 2026

Dear Dr Swaroop,

We are pleased to inform you that your manuscript entitled "Lactate and histone H3K18 lactylation are associated with metabolic control of gene expression in the retina" has been editorially accepted for publication in PLOS Genetics. Congratulations!

The revised manuscript was reviewed and evaluated at the editorial level; overall we think.the revision has addressed the previous concerns.Before your submission can be formally accepted and sent to production you will need to complete our formatting changes, which you will receive in a follow up email. Please be aware that it may take several days for you to receive this email; during this time no action is required by you. Please note: the accept date on your published article will reflect the date of this provisional acceptance, but your manuscript will not be scheduled for publication until the required changes have been made.

Yours sincerely,

Gregory S. Barsh

Academic Editor

PLOS Genetics

John Greally

Section Editor

PLOS Genetics

Aimée Dudley

Editor-in-Chief

PLOS Genetics

Anne Goriely

Editor-in-Chief

PLOS Genetics

BlueSky: @plos.bsky.social

Comments from the reviewers (if applicable):

**Data Deposition**

If you have submitted a Research Article or Front Matter that has associated data that are not suitable for deposition in a subject-specific public repository (such as GenBank or ArrayExpress), one way to make that data available is to deposit it in the Dryad Digital Repository. As you may recall, we ask all authors to agree to make data available; this is one way to achieve that. A full list of recommended repositories can be found on our . As you may recall, we ask all authors to agree to make data available; this is one way to achieve that. A full list of recommended repositories can be found on our . As you may recall, we ask all authors to agree to make data available; this is one way to achieve that. A full list of recommended repositories can be found on our . As you may recall, we ask all authors to agree to make data available; this is one way to achieve that. A full list of recommended repositories can be found on our website....

http://datadryad.org/submit?journalID=pgenetics&manu=PGENETICS-D-25-01094R1

Additionally, please be aware that our data availability policy requires that all numerical data underlying display items are included with the submission, and you will need to provide this before we can formally accept your manuscript, if not already present. requires that all numerical data underlying display items are included with the submission, and you will need to provide this before we can formally accept your manuscript, if not already present. requires that all numerical data underlying display items are included with the submission, and you will need to provide this before we can formally accept your manuscript, if not already present. requires that all numerical data underlying display items are included with the submission, and you will need to provide this before we can formally accept your manuscript, if not already present.

**Press Queries**

If you or your institution will be preparing press materials for this manuscript, or if you need to know your paper's publication date for media purposes, please inform the journal staff as soon as possible so that your submission can be scheduled accordingly. Your manuscript will remain under a strict press embargo until the publication date and time. This means an early version of your manuscript will not be published ahead of your final version. PLOS Genetics may also choose to issue a press release for your article. If there's anything the journal should know or you'd like more information, please get in touch via plosgenetics@plos.org....

---

## [Editor Report · Acceptance letter]

PGENETICS-D-25-01094R1

Lactate and histone H3K18 lactylation are associated with metabolic control of gene expression in the retina

Dear Dr Swaroop,

We are pleased to inform you that your manuscript entitled "Lactate and histone H3K18 lactylation are associated with metabolic control of gene expression in the retina" has been formally accepted for publication in PLOS Genetics! Your manuscript is now with our production department and you will be notified of the publication date in due course.

With kind regards,

Anita Estes

PLOS Genetics

On behalf of:
